# PALB2 chromatin recruitment restores homologous recombination in BRCA1-deficient cells depleted of 53BP1

Rimma Belotserkovskaya [1,2]*, Elisenda Raga Gil[1,2,4], Nicola Lawrence [1], Richard Butler[1], Gillian Clifford[3], Marcus D. Wilson [3]* & Stephen P. Jackson [1,2]*

Loss of functional BRCA1 protein leads to defects in DNA double-strand break (DSB) repair by homologous recombination (HR) and renders cells hypersensitive to poly (ADP-ribose) polymerase (PARP) inhibitors used to treat BRCA1/2-deficient cancers. However, upon chronic treatment of BRCA1-mutant cells with PARP inhibitors, resistant clones can arise via several mechanisms, including loss of 53BP1 or its downstream co-factors. Defects in the 53BP1 axis partially restore the ability of a BRCA1-deficient cell to form RAD51 filaments at resected DSBs in a PALB2- and BRCA2-dependent manner, and thereby repair DSBs by HR. Here we show that depleting 53BP1 in BRCA1-null cells restores PALB2 accrual at resected DSBs. Moreover, we demonstrate that PALB2 DSB recruitment in BRCA1/53BP1-deficient cells is mediated by an interaction between PALB2's chromatin associated motif (ChAM) and the nucleosome acidic patch region, which in 53BP1-expressing cells is bound by 53BP1's ubiquitin-directed recruitment (UDR) domain.

[1] Wellcome Trust CRUK Gurdon Institute, University of Cambridge, Tennis Court Road, Cambridge CB2 1QN, UK. [2] Department of Biochemistry, University of Cambridge, Tennis Court Road, Cambridge CB2 1GA, UK. [3] Wellcome Centre for Cell Biology, University of Edinburgh, Michael Swann Building, Kings Buildings, Mayfield Road, Edinburgh EH9 3JR, UK. [4]Present address: School of Life Sciences, University of Dundee, Dow Street, Dundee DD1 5EH, UK. *email: rb427@cam.ac.uk; marcus.wilson@ed.ac.uk; s.jackson@gurdon.cam.ac.uk

Homologous recombination (HR) is largely an error-free pathway in which the sister chromatid is used as a template for repairing genetic information in the damaged DNA molecule. For HR to ensue, a tract of single-stranded DNA (ssDNA) with a 3′ overhang is first generated by the concerted actions of endo- and exonucleases such as MRE11, CtIP, EXO1 and DNA2 in a process called DNA end-resection[1]. Resected DNA is rapidly coated with replication protein A (RPA) multimers that protect ssDNA from both nucleolytic degradation and self-hybridisation to form complex secondary structures. Subsequently, RPA is replaced by the RAD51 recombinase protein to form a nucleoprotein filament competent for invasion of the homologous DNA template. This step is promoted by BRCA2, which binds several molecules of RAD51 delivering them to the RPA-coated ssDNA and, together with other factors, facilitates replacement of RPA with RAD51[2,3].

BRCA1 promotes HR in at least two functionally and temporally distinct stages: first, by modulating the kinetics of DNA end-resection[4,5] and later by promoting RAD51 filament formation via recruiting the PALB2-BRCA2-RAD51 complex to RPA-coated ssDNA[6,7]. Interfering with either of these steps results in HR impairment, a key hallmark of BRCA1 deficiency. As loss of 53BP1 or its downstream effectors partially rescues the HR defect of cells lacking functional BRCA1[8,9], an important unresolved question is how this is achieved: does it act by rebalancing DNA end-resection, by restoring proficient RAD51 loading, or both?

Herein, we employed a combination of cell biology and biochemical approaches to understand the recruitment of PALB2 to sites of DNA DSBs. We find that in the absence of BRCA1, loss of 53BP1 creates a permissive environment for PALB2 recruitment. We show that in BRCA1/53BP1-deficient cells, PALB2 accumulation into DNA-damage-induced foci requires the intact Chromatin Association Motif (ChAM), which mediates a direct interaction with the acidic patch of the nucleosome. Our findings elucidate the role of PALB2 in promoting HR and help to explain how DNA repair can be restored in clinically relevant BRCA1/53BP1-deficient cells.

## Results

### 53BP1 loss restores HR in BRCA1- but not PALB2-depleted cells

To address how HR can be restored in the absence of BRCA1, we characterised the impacts of 53BP1 on various stages of HR in cells depleted for BRCA1, PALB2 or BRCA2. In gene conversion assays using the traffic light reporter (TLR) system integrated into human U2OS cells[10,11], short-interfering RNA (siRNA)-mediated depletion of BRCA1, PALB2 or BRCA2 severely inhibited HR (Fig. 1a, b). Furthermore, consistent with previous studies[8,9], 53BP1 depletion partially rescued the HR defect of BRCA1-depleted cells. Notably however, 53BP1 depletion did not rescue the HR defect of PALB2- or BRCA2-depleted cells (Fig. 1a; the latter is in agreement with the observations that 53BP1 loss did not rescue viability of $Brca2^{-/-}$ mouse cells[9] or the HR defect of Palb2-deficient mouse cells[12]). Nevertheless, while 53BP1 depletion consistently enhanced HR up to threefold in the BRCA1-depleted background, HR never exceeded 30% of control levels. To ascertain whether such inefficient HR rescue was at least in part due to incomplete 53BP1 depletion, we performed HR assays in U2OS-TLR cells engineered to be 53BP1 gene knock-outs (KOs) by means of CRISPR-Cas9 genome editing (Fig. 1c–e). While BRCA1 depletion markedly reduced HR in U2OS-TLR cells containing wild-type (WT) 53BP1, its depletion in 53BP1 KO backgrounds resulted in a considerably less pronounced HR defect (Fig. 1c). By contrast, depletion of PALB2 almost completely abrogated HR in both 53BP1-proficient and 53BP1 KO cells (Fig. 1d). Taken together with our other data, these findings indicated that 53BP1 loss

suppresses the HR defect caused by BRCA1 deficiency but not that caused by PALB2 deficiency.

HR proficiency correlates with the ability of a cell to load the RAD51 recombinase on DNA, which can be detected in immunofluorescence studies as nuclear RAD51 foci[13]. Consistent with our data from the HR reporter assays were our findings from microscopy experiments in which we quantified RAD51 foci in cyclin A-positive (to mark S- and G2-phase cells, where HR is preferentially active) RPE1 cells depleted for various proteins and subjected to ionising radiation (IR) to induce DNA damage. Thus, while depletion of either BRCA1 or PALB2 resulted in a clear defect in RAD51 IR-induced focus (IRIF) formation, only in the case of BRCA1-depleted cells was this defect rescued by downregulation of 53BP1 (Fig. 1f; as shown in Supplementary Fig. 1, when we carried out studies with RPE1 cells knocked out for BRCA1 and/or 53BP1, this also revealed that the RAD51 IRIF formation defect caused by BRCA1 loss was rescued by 53BP1 inactivation).

Although BRCA1 is pivotal to normal regulation of the kinetics of DSB resection in actively replicating cells exposed to agents such as DNA restriction enzymes, DNA topoisomerase inhibitors or PARP1 inhibitors, the protein itself is not required for the actual process of resection but is important to counteract inhibition of resection exerted by 53BP1 and its downstream effectors (reviewed in ref.[5]). Accordingly, DNA end-resection in BRCA1-deficient cells occurs at a slower rate than in BRCA1-proficient control cells[4,14–16]. Previous studies in which physical lengths of ssDNA were measured via the Single Molecule Analysis of Resection Tracks (SMART) approach[4,17] demonstrated that although ssDNA tracks were ~1.5 times shorter in BRCA1-deficient cells than in control cells, they still encompassed tens of microns of DNA. In agreement with these observations, we consistently found by flow cytometry- and confocal microscopy-based methods, either very minor or no significant effect of BRCA1 loss on ssDNA or RPA focus formation in S/G2-phase cells treated with IR or the topoisomerase 1 inhibitor, camptothecin (Supplementary Fig. 1f). Nevertheless, as mentioned above, RAD51 IRIF formation was severely affected in BRCA1-deficient cells, even at late time points (up to 8 h after irradiation). Taken together with previous findings[18,19], our data thus indicated that BRCA1 is critical for RAD51 filament formation and ensuing HR. In light of this, we concluded that the pronounced effects of BRCA1 loss on RAD51 filament formation and HR might largely reflect BRCA1's crucial role(s) associated with recruitment of the PALB2-BRCA2-RAD51 complex to RPA-coated ssDNA[6,7,20].

### PALB2 IRIF restoration by 53BP1 loss in BRCA1-negative cells

Given the above results and our observation that 53BP1 depletion rescued both the RAD51 loading and HR defects of BRCA1-deficient but not PALB2-deficient cells, we tested whether 53BP1 depletion alleviated impaired PALB2 DNA-damage recruitment in BRCA1-deficient cells. Using RPE1 cells in which the endogenous PALB2 gene was tagged with the green fluorescent protein (GFP) variant Venus (Supplementary Fig. 2a–g), we observed that 53BP1 depletion indeed rescued the defect of BRCA1-depleted cells in mediating PALB2 recruitment to regions containing RPA-coated, resected DSBs (Fig. 2a, b and Supplementary Fig. 2h). This was also true for untagged PALB2, assayed by using an antibody against endogenous PALB2[21] to probe RPE1 cells depleted for BRCA1 or both BRCA1 and 53BP1 (Supplementary Fig. 3a, b). Furthermore, similar results were obtained when we examined recruitment of GFP-PALB2 to DNA-damage tracks generated by laser micro-irradiation of U2OS cells (Supplementary Fig. 3c, d).

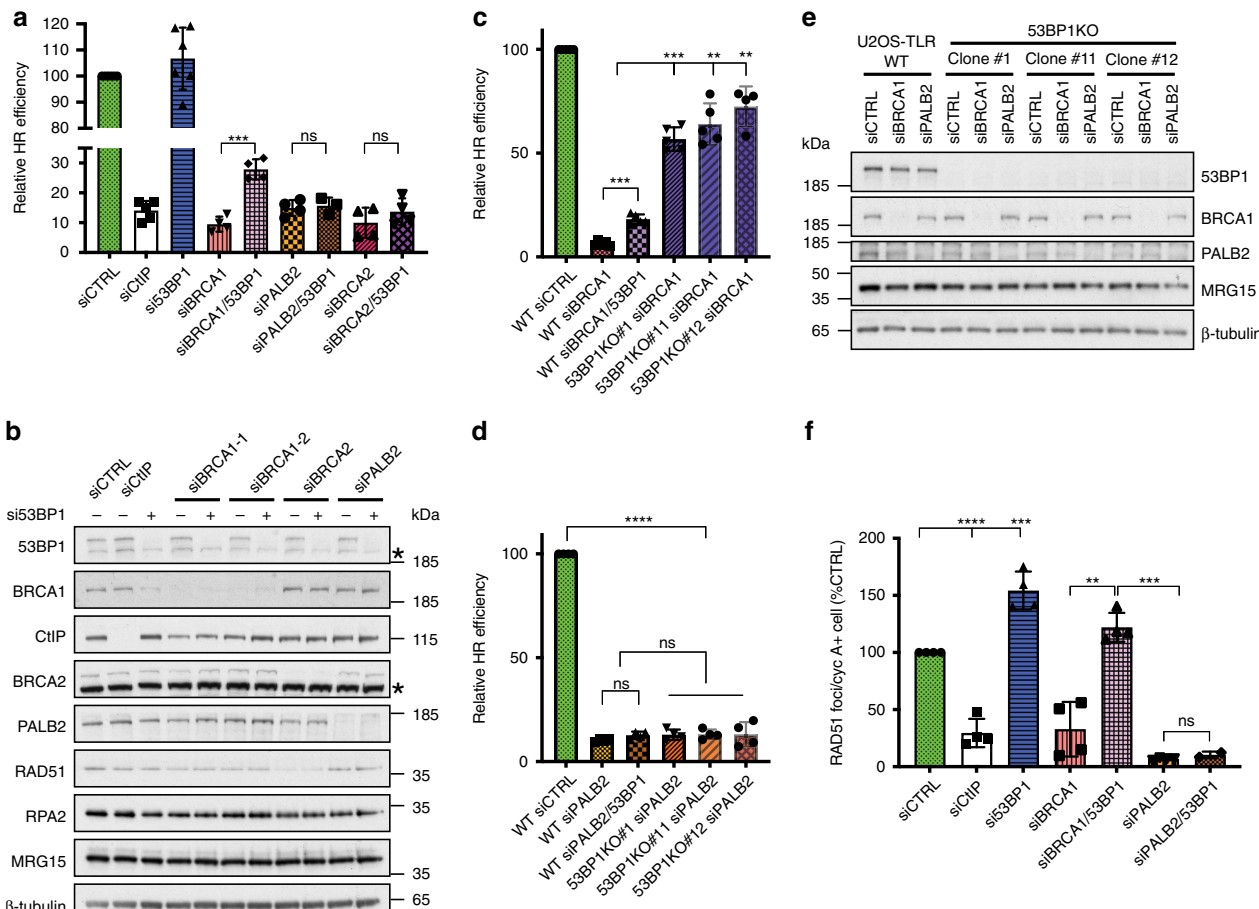

**Fig. 1 53BP1 loss corrects HR in BRCA1- but not in PALB2- or BRCA2-deficient cells. a** HR reporter assay in U2OS-TLR WT cells siRNA-depleted for indicated proteins or treated with a control siRNA (siCTRL). The bars represent mean ± st.dev.; unpaired *t* test analyses were conducted to determine if differences between samples were statistically significant; $n \geq 3$ experiments. Individual data points are plotted over bars. **b** Immunoblot of extracts of U2OS-TLR cells depleted for indicated proteins and used in HR assays shown in (**a**). Asterisks mark nonspecific bands. **c, d** Comparison of HR efficiencies measured by TLR assays performed in U2OS-TLR WT and *53BP1* KO cells siRNA-depleted for either BRCA1 (**c**) or PALB2 (**d**). Data representation and statistical analyses are as in (**a**); $n \geq 4$ experiments. **e** Immunoblot of extracts of U2OS-TLR WT and *53BP1* KO cells siRNA-depleted for BRCA1 and PALB2 and used in HR assays in (**c, d**). **f** Quantification of RAD51 IRIF in RPE1 cells siRNA-depleted for indicated proteins. Cells were treated with 6 Gy of IR, fixed at 4−8 h after irradiation, stained with antibodies specific to cyclin A and RAD51 proteins, imaged and quantified using OPERA Phoenix HT microscope; $n = 4$ repeats. Source data are provided as a Source Data file. **$P < 0.01$, ***$P < 0.001$, ****$P < 0.0001$; ns, not significant ($P \geq 0.05$).

While carrying out our studies, we noticed that, upon 53BP1 depletion, PALB2 tends to form not only more numerous but also more discernible foci (Fig. 2a, b and Supplementary Fig. 3a), as well as brighter lines at laser tracks (Supplementary Fig. 3c, d), which could potentially be explained by accumulation of more PALB2 molecules at DSBs. To assess this possibility, we used higher resolution imaging employing three-dimensional structured illumination microscopy (3D-SIM)[22], which allowed us to estimate the number of PALB2 IRIF juxtaposed to individual RPA fibres in the nucleus. Thus, we found that 53BP1 depletion led to an increase in the average number of PALB2 foci adjacent to each RPA focus (Fig. 2c; Supplementary Movies). To our knowledge, this is the first direct demonstration that 53BP1 loss both restores PALB2 IRIF formation in BRCA1-deficient cells and enhances PALB2 IRIF intensity in BRCA1-proficient cells. Together, these results suggested that 53BP1 might act to directly or indirectly interfere with PALB2 recruitment to sites containing resected DSBs.

**ChAM supports PALB2 IRIF formation in BRCA1/53BP1-deficient cells.** In light of our hypothesis that 53BP1 might

interfere with PALB2 accumulation at DSB sites, and considering that both 53BP1 and PALB2 bind chromatin, we decided to test whether it is the ability of PALB2 to interact with chromatin that supports its accumulation at DSB sites in the absence of BRCA1 and 53BP1. The chromatin-binding modes of 53BP1 are well established: its tandem Tudor domain interacts with non-damage specific histone H4 methylated on lysine 20[23], while its ubiquitin-directed recruitment motif (UDR) is critical for 53BP1 accumulation at DSB sites following RNF168-dependent ubiquitylation of histone H2A on lysine 15 ($H2A^{K15ub}$)[24]. In addition to its interaction with $H2A^{K15ub}$, the 53BP1 UDR also contacts the nucleosome surface, including interactions with the acidic patch formed by negatively charged residues in histone H2A-H2B within the nucleosome[25]. By contrast, chromatin-binding by PALB2 is modulated by a short region termed the Chromatin Associated Motif (ChAM)[26], and PALB2 can also be recruited to chromatin by its interacting partner MRG15, a protein found in several chromatin-modifying and chromatin-remodelling complexes[27,28]. Thus, we assessed how efficiently GFP-PALB2 derivatives lacking either the ChAM or the MRG15-interacting domain, or both (Fig. 3a and Supplementary Fig. 4), formed IRIF in the presence or absence of 53BP1 and BRCA1. As

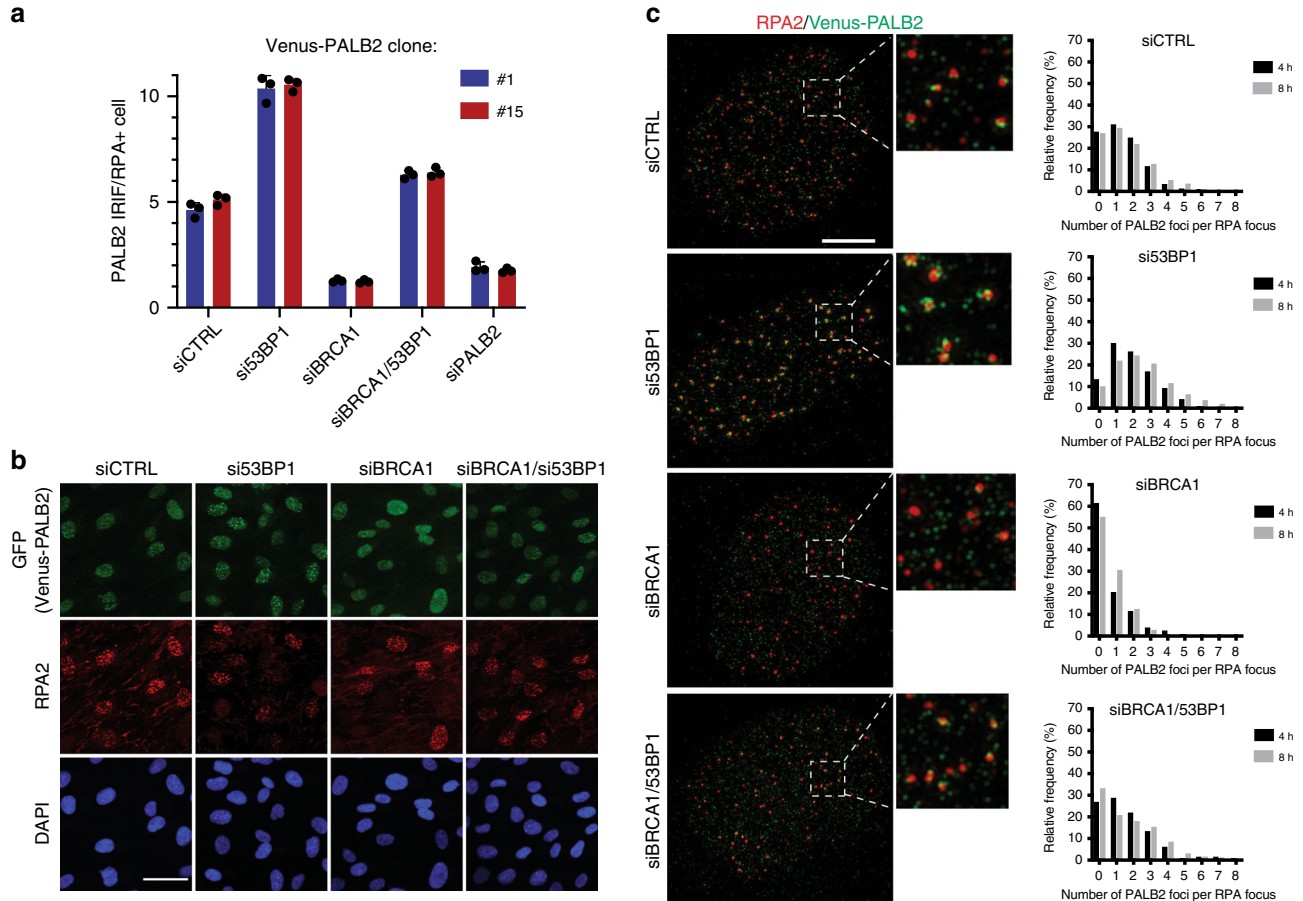

**Fig. 2 53BP1 depletion rescues PALB2 focus formation in BRCA1-deficient cells. a** Quantification of Venus-PALB2 IRIF in RPA focus-positive RPE1 cells. Two independently generated RPE1 Venus-PALB2 cell lines (#1 and #15) were siRNA-depleted for indicated proteins, exposed to 6 Gy of IR and 6 h later, fixed and stained with anti-GFP and anti-RPA2 antibodies. Imaging and IRIF quantifications were performed in three independent experiments, using OPERA Phoenix HT microscope. **b** Representative images, acquired on OPERA Phoenix HT microscope, of RPE1 cells with endogenously Venus-tagged *PALB2* gene. The cells were stained with anti-GFP (to enhance the signal of the Venus tag) and anti-RPA2 antibodies. Scale bar, 50 μm. **c** Venus-PALB2 association with RPA filaments in cells depleted for 53BP1. RPE1 cells expressing endogenously tagged Venus-PALB2 were depleted for BRCA1 and/or 53BP1, irradiated with 6 Gy of IR and, 8 h later, processed for immunofluorescence analyses. Images were acquired using super-resolution 3D-SIM OMX microscope. Scale bar, 5 μm. Graphs to the right of the images represent distribution of relative frequencies of Venus-PALB2 foci numbers adjacent to each RPA focus. Source data are provided as a Source Data file.

expected[26,29], in BRCA1/53BP1-proficient (WT) RPE1 cells, deletion of either the ChAM or MRG15-binding domain had a modest effect on PALB2 IRIF formation, whereas lack of both regions had a more pronounced effect (Fig. 3b, c and Supplementary Fig. 4a). In stark contrast, in the absence of both BRCA1 and 53BP1, PALB2 recruitment to IRIF was dramatically impaired by deletion of its ChAM domain and was also diminished somewhat upon deletion of the MRG15-interacting region (Fig. 3d, e and Supplementary Fig. 4b). Notably, in *BRCA1/53BP1* double KO cells, a PALB2 derivative lacking both ChAM and MRG15-binding domains had an IRIF formation defect no worse than that of the single ChAM deletion mutant (Fig. 3d, e) highlighting the key importance of the ChAM domain in this particular genetic background. Based on these findings, we concluded that while the main recruitment mode for PALB2 to DSB sites is normally via its physical interaction with BRCA1, in the absence of BRCA1, PALB2 accumulation at DSBs strongly depends on its intrinsic chromatin-binding properties mediated by the ChAM region.

To assess the biological relevance of these observations and conclusions, we examined impacts on cellular sensitivity to the PARP1 inhibitor olaparib, which induces DNA damage that is normally repaired by HR mechanisms in S-phase[26,30]. In agreement with our other observations, complementation of PALB2-depleted RPE1 cells with PALB2$^{\Delta ChAM}$ did not confer significant olaparib sensitivity when BRCA1 was present (Fig. 3f). By contrast, when parallel studies were performed with cells lacking BRCA1 and 53BP1, wild-type PALB2 imparted olaparib resistance while PALB2$^{\Delta ChAM}$ did not (Fig. 3g). Collectively, these results indicated that when PALB2 cannot be recruited to DNA-damage sites by BRCA1, 53BP1 functions to antagonise PALB2 binding to DNA lesion-proximal chromatin and thereby impairs HR. They also suggested that, when BRCA1-deficient cells also lack 53BP1, PALB2 is recruited to DSB regions where it forms IRIF in a manner dependent on its ChAM region and promotes HR.

**PALB2 ChAM domain contacts the nucleosome acidic patch.** To complement our cell-based studies and to further investigate the chromatin-binding properties of the PALB2 ChAM region, we characterised the ability of bacterially expressed and purified 6xHis-MBP-tagged PALB2 ChAM protein to interact with defined recombinant nucleosome core particles (NCPs) in vitro.

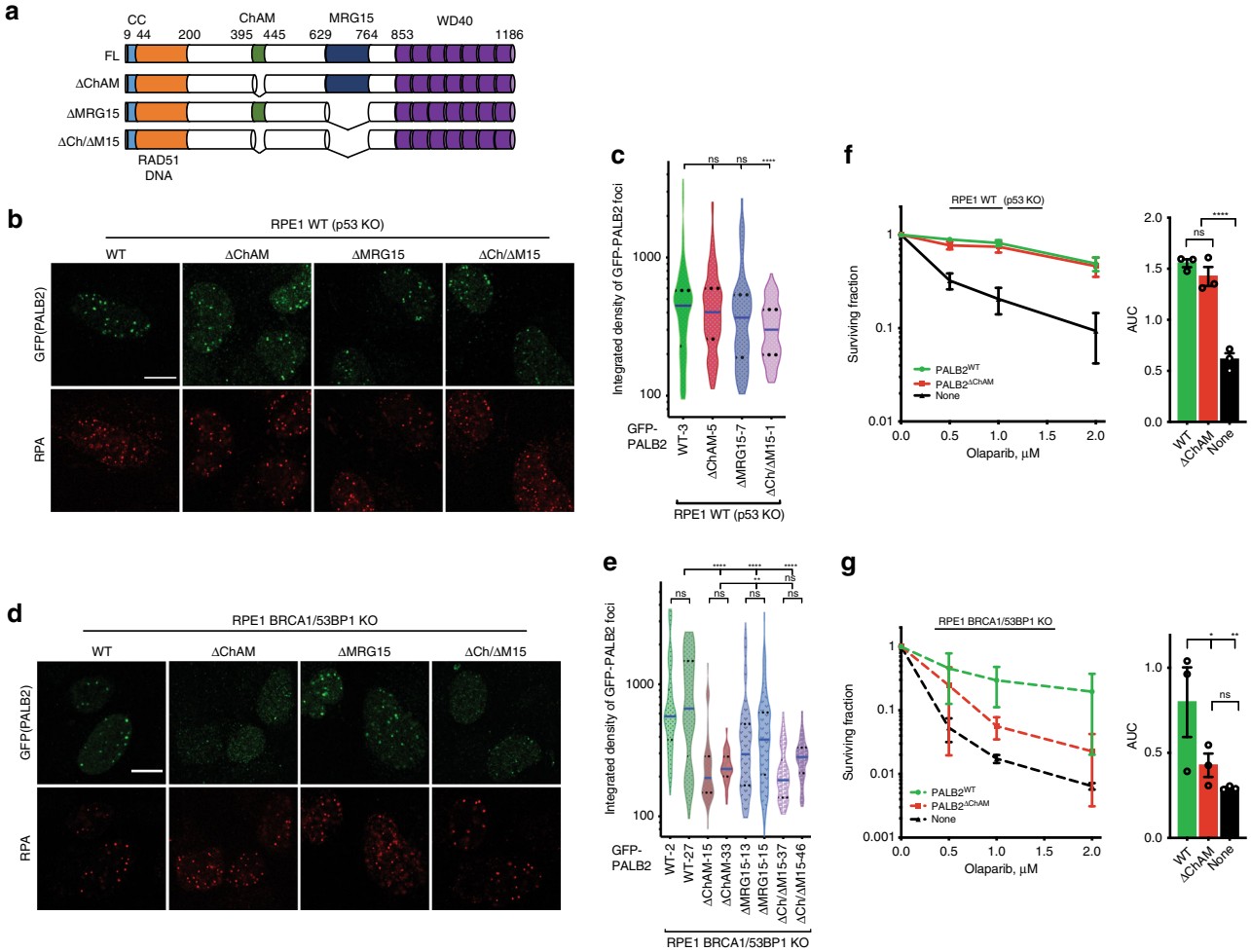

**Fig. 3 ChAM of PALB2 is important for PALB2 recruitment to DSB sites in BRCA1/53BP1-deficient cells. a** Domain structure of GFP-tagged derivatives of PALB2 stably integrated at the FRT site in Tet repressor-expressing RPE1 cells. **b** Representative images of RPE1 *p53* KO cells siRNA-depleted of endogenous PALB2 and complemented with indicated GFP-PALB2 variants. Cells were exposed to 6 Gy of IR and 6 h later fixed and stained with anti-GFP (to detect PALB2) and anti-RPA antibodies. Scale bars, 10 μm. Additional images are in Supplementary Fig. 4a. **c** Violin plots represent quantification of the integrated density of GFP-PALB2 IRIF in RPA-positive cells following exposure to 6 Gy of IR. Numbers next to the names of the cell lines indicate numbers of the individual clones used in the experiment. **d** Representative images of RPE1 *p53/BRCA1/53BP1* KO cells siRNA-depleted of endogenous PALB2 and complemented with indicated GFP-PALB2 variants. Cells were treated as in (**a**). Scale bars, 10 μm. Additional images are in Supplementary Fig. 4b. **e** Violin plots represent quantification of the integrated density of GFP-PALB2 IRIF in RPE1 p53/BRCA1/53BP1 cells following exposure to 6 Gy of IR. Numbers next to the names of the cell lines indicate numbers of the individual clones used in the experiment. **f, g** Clonogenic survivals of RPE1 *p53* KO (**f**) or *p53/BRCA1/53BP1* KO (**g**) cells complemented with either PALB2^WT or PALB2^ΔChAM in response to indicated doses of olaparib. Lower panels show area under curve (AUC); n ≥ 3. The bars represent mean ± s.e.m.; one-way ANOVA; *P < 0.05, **P < 0.01, ****P < 0.0001; ns, not significant (P ≥ 0.05); n = 3 independent experiments. Source data are provided as a Source Data file.

Thus, we found that the ~50 amino acid residue ChAM domain was sufficient to retrieve NCPs in biochemical pull-down assays (Fig. 4a). Furthermore, removing the N-terminal tails of either histone H3 or H4 did not impair ChAM binding, suggesting that the interaction may be mediated by the core nucleosome surface. Having considered that the ChAM domain is highly positively charged (with a predicted pI of 9.6; ExPasy) and that the nucleosome acidic patch is a common site of contact for chromatin-binding proteins, including 53BP1[31,32], we tested whether the interaction between ChAM and NCPs was affected when the negative charge of the H2A-H2B acidic patch (Fig. 4b) was reduced by mutating key residues (H2A Glu61, 90, 91 and H2B Glu105 to Ala). Notably, we found that mutating the acidic patch ablated ChAM domain binding to NCPs (Fig. 4a), implying that the ChAM–nucleosome interaction is at least in part mediated by the acidic patch. As PALB2 IRIF formation appears to be dependent on RNF168[33,34], which is recruited to DSBs and

ubiquitylates H2A at Lys15, we tested if histone H2A Lys15 ubiquitylation affected PALB2 ChAM binding in vitro and found that both unmodified and H2A Lys15-ubiquitylated NCPs bound ChAM comparably (Supplementary Fig. 5a). Collectively, our data indicated that the PALB2 ChAM binds chromatin independently of H2A Lys15 ubiquitylation status, but binding requires an intact H2A-H2B acidic patch.

Reasoning that the nucleosome acidic patch might be recognised at least in part via electrostatic interactions with positively charged amino acid residues in the PALB2 ChAM region, we mutated four conserved basic residues in the ChAM domain (Fig. 4c), and found that these mutations resulted in considerable attenuation of the ChAM-nucleosome interaction (Supplementary Fig. 5b). In agreement with our in vitro binding experiments, cells complemented with PALB2 containing these four mutations (PALB2^4M; Fig. 4c) exhibited properties similar to those of PALB2^ΔChAM (Fig. 4d, e and Supplementary Fig. 5c).

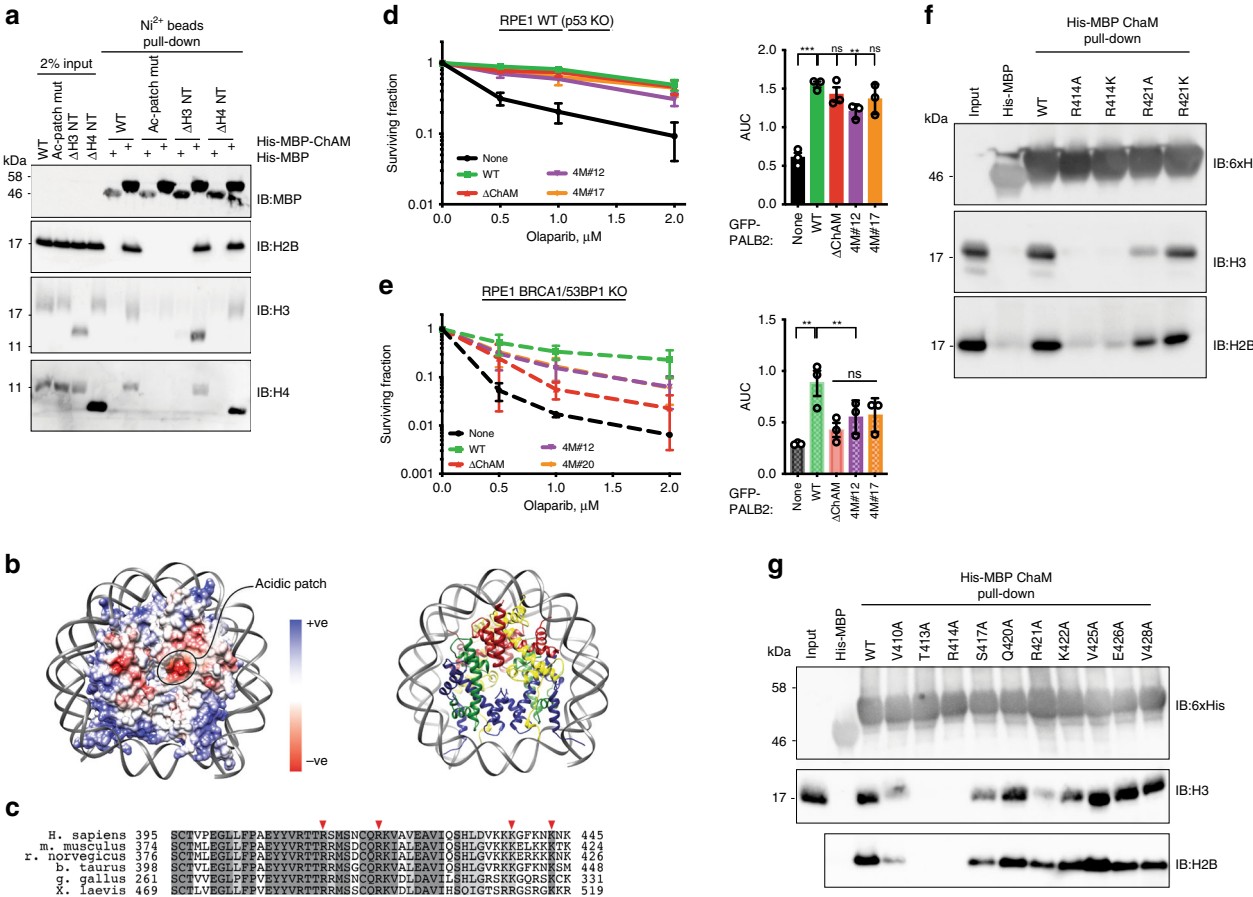

**Fig. 4 ChAM domain of PALB2 interacts with nucleosome acidic patch. a** In vitro pull-down (PD) assay using purified components. 6xHis-MBP-tagged PALB2 ChAM domain or 6xHis-MBP immobilised on Ni-NTA affinity beads, incubated with recombinant nucleosome variants. Acidic patch mutant (Ac-patch mut) comprises H2A E61/91/92A and H2B E105A; ΔH3 NT comprises H3 residues 25−135; ΔH4 NT comprises H4 residues 21−106. Anti-H3 and H4 antibodies showed increased reactivity towards histones with shorter N-terminal tails. **b** The nucleosome and its acidic patch. Left panel: electrostatic potential view of the protein histone surface; right panel: ribbon representation of the nucleosome core particle. **c** Sequence alignment of ChAM with alanine scanning mutation areas highlighted (blue line) and basic residues mutated in 4 M mutant highlighted (red arrows). **d**, **e** Clonogenic survivals of RPE1 p53 KO (**e**) or p53/BRCA1/53BP1 KO (**f**) cells complemented with PALB2$^{WT}$, PALB2$^{ΔChAM}$ or PALB2$^{4M}$ in response to indicated doses of olaparib. Lower panels show corresponding area under curve (AUC) bar graphs; $n ≥ 3$. The bars represent mean ± s.e.m.; one-way ANOVA; **$P < 0.01$, ***$P < 0.001$; ns, not significant ($P ≥ 0.05$); $n = 3$ independent experiments. Source data are provided as a Source Data file. **f** Pull-down assays comparing NCP association of 6xHis-MBP-PALB2 ChAM variants with charge removal and charge retaining ChAM mutations on Arg414 and Arg421. **g** Pull-down assays between NCPs and immobilised 6xHis-MBP-ChAM variants containing alanine mutations across the proposed nucleosome-interacting region.

Thus, while PALB2$^{ΔALB2}$ and PALB2$^{4M}$ both largely alleviated the olaparib hypersensitivity of BRCA1/53BP1-proficient cells depleted of endogenous PALB2 protein (Fig. 4d), neither achieved this in the *BRCA1/53BP1* double knock-out background (Fig. 4e).

To gain further insight into the nature of the ChAM−NCP interaction, we tested whether it was affected by mutating basic residues Arg414 and Arg421 to either alanine (to remove the charge) or lysine (to preserve the positive charge). We found that both R414A and R414K mutants had lost their ability to bind NCPs (Fig. 4f), suggesting that Arg414 may play a critical structural role in the ChAM−NCP interaction. By contrast, while the R421A mutation reduced ChAM−NCP binding, the R421K mutant retained NCP binding potential (Fig. 4f), suggesting that this residue functions primarily via forming electrostatic interactions with the negatively charged H2A-H2B acidic patch.

In further biochemical pull-down studies, we explored potential contributions to NCP binding of several other residues within the highly conserved core of the PALB2 ChAM region. Thus, we found that alteration of several residues centred around Arg414 (mutations V410A, T413A and S417A) compromised the interaction with recombinant NCPs (Fig. 4g). Interestingly,

cancer-associated mutations in this region have been previously reported to exhibit reduced chromatin association in chromatin fractionation studies[29]. In line with this, we observed that reported missense mutations[35] T413S, R414Q and S417Y, but not V425M, displayed reduced interaction with NCPs in vitro (Supplementary Fig. 5d).

## Discussion

The key hallmark of BRCA1-deficient cells is their inability to repair DNA damage via HR. Such an impairment can be explained by the pivotal roles BRCA1 plays in two major steps of HR: firstly, it stimulates end-resection of DSBs, thus fostering the generation of ssDNA required for HR-mediated homology search; secondly, BRCA1 promotes recruitment of PALB2 and BRCA2 to DNA-damage sites, which in turn help load RAD51 recombinase onto resected ssDNA (reviewed in ref. [36]). When it was first demonstrated that loss of 53BP1 corrects HR defects of BRCA1-deficient cells and suppresses lethality of *Brca1*$^{−/−}$ mice, it was postulated that restoration of DNA end-resection was the key event mediating these impacts[8,9]. While this served as a useful

starting model, further extensive research, which includes a very recent study describing roles for 53BP1 both pre- and post-DNA end-resection[37], has unravelled a much more complex relationship between BRCA1, 53BP1 and the processes of HR (reviewed in refs. [5,36]). In agreement with several studies showing that DNA end-resection takes place in BRCA1-deficient cells, albeit with delayed kinetics[4,14–16], we could detect similar amounts of chromatinised RPA in BRCA1-proficient and -deficient cells that had been treated with DSB-inducing agents and allowed to process the lesions for several hours. However, even at the late time points, BRCA1-deficient cells failed to form RAD51 IRIF unless they were co-depleted for 53BP1. This strongly suggested that, in addition to slowing down initial DSB resection, 53BP1 could also interfere with RAD51 loading in BRCA1-deficient cells. Thus, we set out to gain a better mechanistic understanding of how 53BP1 loss restores RAD51 recruitment in BRCA1-deficient cells.

Our results support a model in which RAD51 nucleoprotein filament formation and HR are restored in BRCA1-deficient cells following 53BP1 loss via a mechanism underpinned by recovery of PALB2 recruitment to DSB-proximal chromatin mediated by a direct interaction between the PALB2 ChAM and the nucleosome acidic patch region. In this model, in the absence of BRCA1, 53BP1 is not efficiently displaced from chromatin adjacent to DSBs[17,38] and therefore remains bound to nucleosomes via both its Tudor and UDR domains, with the latter helping to occlude the H2A-H2B acidic patch, which is recognised by various chromatin-interacting proteins[32]. We propose that, upon loss of 53BP1, the surfaces on nucleosomes where it binds, including the acidic patch, become more available for interaction with other less abundant or not as avidly binding factors, including PALB2.

Many chromatin-binding proteins described to date interact with nucleosomes via the acidic patch[31,39]. A common structural arrangement is the insertion of an arginine anchor into the acidic patch, with stabilising interactions formed by other positively charged residues. Interestingly, in the conserved core of the ChAM domain, we find two arginine residues, Arg414 and Arg421, mutations of which reduce the ChAM interaction with nucleosomes. Intriguingly, while the ChAM–NCP interaction takes place in the context of the conservative R421K substitution, NCP binding was ablated by changing Arg414 to alanine, lysine or glutamine (the latter is a mutation found in a cancer patient[35]). This suggests that the structure of Arg414 itself, rather than simply its electrostatic charge, plays a key role in binding to the nucleosome. Further work is required to determine the exact nature of ChAM binding to nucleosomes, whether the interaction is solely focused on the acidic patch, and if Arg414 is indeed operating as an arginine anchor.

While the ChAM domain is sufficient for interaction with recombinant NCPs in vitro, it is likely that PALB2 chromatin recruitment in cells is multivalent in nature. Most chromatin-binding proteins/complexes described to date recognise multiple interfaces to ensure both affinity and specificity of their recruitment to the correct subcellular location[32]. PALB2 binds to nucleosomes in our in vitro assays with low affinity, requiring large amounts of immobilised PALB2. In vivo, other PALB2 interaction interfaces, including direct interaction with DNA[40] and other DDR factors[33,34,41,42], likely ensure appropriate PALB2 recruitment to sites of DSBs. Our data support a model in which PALB2's interaction with BRCA1 plays a dominant role in PALB2 recruitment to DNA-damage sites under normal circumstances, while when BRCA1 and 53BP1 are absent, normally stabilising interactions, such as between the ChAM domain and the nucleosome acidic patch, are likely to become more essential for DSB recruitment.

Clearly, if PALB2 recruitment in BRCA1/53BP1-deficient cells were mediated only by its interaction with the H2A-H2B acidic patch, it would be expected to bind chromatin throughout the whole nucleus; however, following DSB induction, PALB2 forms foci adjacent to RPA-coated ssDNA. Such targeted accumulation is probably mediated by the interaction with other DSB recruited factors or recognition of induced histone post-translational modifications. In line with this hypothesis, it has been reported that there is a direct interaction between RNF168 and PALB2[33], and we and others have observed that loss of RNF168 attenuates IRIF formation by PALB2 and re-sensitises BRCA1/53BP1-deficient cells to PARP inhibitors[34] (Supplementary Fig. 6a−d).

One consideration important to our model is that, when 53BP1 is recruited to DSB-proximal chromatin, it will localise there together with components of the recently described Shieldin complex, with 53BP1-RIF1 bound to chromatin and REV7-SHLD1/2/3 contacting the ssDNA of partially resected DSBs[43]. As 53BP1 works together with the rest of its downstream co-factors to block over-resection of DNA DSBs, loss of any of these proteins partially restores RAD51 filament formation, functional HR and PARP-inhibitor resistance in BRCA1-deficient cells. Notably, whereas recruitment of RIF1, REV7 and SHLD1/2/3 depends on intact 53BP1, losing the downstream proteins does not prevent 53BP1 from forming IRIF[44–47]. Thus, in the absence of its co-factors, 53BP1 is still bound to DSB-proximal chromatin and therefore might interfere with efficient PALB2 recruitment, especially in BRCA1-deficient settings. Importantly, according to a recent study that employed super-resolution microscopy to characterise chromatin conformations orchestrated by 53BP1 and its co-factors, 53BP1 accumulates near DSB sites and, together with RIF1, forms topologically associating domains (TADs) of compacted chromatin[47]. Whereas the integrity of these TADs is fully dependent on 53BP1 and is partially compromised by RIF1 depletion, loss of Shieldin components does not affect 53BP1/RIF1-mediated chromatin compaction[47]. In line with these observations and our model that loss of 53BP1 in BRCA1-deficient cells creates permissive conditions for PALB2 accumulation on chromatin, we see that only co-depletion of 53BP1 fully rescues Venus-PALB2 and RAD51 foci in BRCA1-depleted RPE1 cells, while RIF1 depletion has a weaker effect and the loss of REV7 or Shieldin components does not allow detectable recruitment of PALB2 to DSB-proximal chromatin (Supplementary Fig. 7). Our findings are in agreement with studies demonstrating that losing 53BP1 has the strongest overall effect in rescuing BRCA1-deficient phenotypes[45,48,49]. Accordingly, we propose that upon depletion of RIF1, REV7 or SHLD1/2/3 in BRCA1-deficient cells, PALB2 accumulation into IRIF on DSB-proximal chromatin is attenuated by 53BP1 but under such circumstances, partial rescue of HR is brought about by Shieldin loss allowing some PALB2/RAD51 loading directly onto ssDNA generated at DSB sites. Consistent with this notion are findings suggesting that Shieldin might interfere with RNF168-mediated PALB2 recruitment to resected DNA[37]. Further studies are needed to define the precise mechanisms underpinning these effects, and to establish whether and how they may relate to PARP-inhibitor resistance in clinical settings.

## Methods

**Cell culture**. U2OS and HEK293 cell lines were originally obtained from the ATCC cell repository and routinely tested to be mycoplasma-free. RPE1 FRT/TR cells were described in ref. [50]. All cells were maintained at 37 °C in a humidified atmosphere containing 5% $CO_2$. U2OS, U2OS-derived and HEK293T cells were cultured in standard Dulbecco modified Eagle's minimal essential medium (DMEM) (Sigma-Aldrich). RPE1 and RPE1-derived cells were cultured in Ham's Nutrient Mixture F12 medium (Sigma-Aldrich) supplemented with 17.5 ml of 7.5% NaHCO3 solution per 500 ml (Sigma-Aldrich). All media were supplemented with 10% fetal bovine serum (FBS; BioSera), 2 mM L-glutamine, 100 U ml⁻¹ penicillin, and 100 μg ml⁻¹ streptomycin (Sigma-Aldrich). Tetracycline-negative FBS from PAA Laboratories was used for U2OS cells stably expressing tetracycline repressor (U2OS-Trex) and derived inducible stable cell lines. Additional supplements used

to maintain stable cell lines were as follows: 2 µg ml⁻¹ puromycin (Sigma-Aldrich) for U2OS TLR cells; 2 µg ml⁻¹ blasticidin (Invitrogen) for U2OS-Trex cells; 0.2 mg ml⁻¹ zeocin (Life technologies) and 2 µg ml⁻¹ blasticidin (Invitrogen) to select U2OS-Trex cells stably expressing inducible EGFP-PALB2; 0.6 mg ml⁻¹ G418 (Invitrogen) to select RPE1 FRT/TR cells with stable inducible expression of EGFP-PALB2 derivatives. To induce EGFP-PALB2 expression, doxycycline (Sigma-Aldrich) was added at a final concentration of 0.1– 0.01 µg ml⁻¹.

**Plasmids and cloning.** EYFP-PALB2 was subcloned from pEYFP-PALB2WT plasmid[26] (a gift from F. Esashi) into pcDNA4-TO (Thermo Fisher Scientific) and pcDNA5-FRT/TR-neo plasmids (pcDNA5/FRT/TO-neo (1795) was a gift from Jonathon Pines (Addgene plasmid # 41000; http://n2t.net/addgene:41000; RRID: Addgene_41000)). PALB2 derivatives lacking ChAM domain (residues 395−445), MRG15-binding domain (residues 629−764) or both were generated by fusion PCR[51]. Details of primers used for cloning and mutagenesis could be provided upon request. PALB2 ChAM sequence (residues 395−445) was optimised for codon usage (R431 AGG-CGT) and inserted into pET 6xHis MBP TEV LIC cloning vector (addgene gift of Scott Gradia) via ligation independent cloning. Point mutations were introduced using QuickChange Lightning Site-Directed Mutagenesis kit (Agilent).

**siRNA and plasmid transfections.** Unless the sequences for siRNA oligonucleotides had come from previous publications, siRNA oligonucleotides were designed using either the MWG (Erofins) or Dharmacon design centre and purchased from MWG Biotech (Erofins). siRNA transfections were performed using Lipofectamine RNAiMAX (Life Technologies) at a final siRNA concentration of ∼50 nM according to the manufacturer's instructions. siRNA-transfected cells were re-plated for assays 48 h after transfection and treated and processed 24 h after re-plating (72 h after transfection). DNA transfections were performed using TransIT-LT1 (Mirus Bio) following the manufacturer's guidelines. As a negative control, we used siRNA oligonucleotides targeting Luciferase (CTRL siRNA). For a complete list of siRNA sequences used in this study, see Supplementary Table 1.

**Stable cell line generation.** U2OS-Trex cells stably expressing inducible EYFP-PALB2 derivatives were generated by transfection of pcDNA4/TO/EYFP-PALB2 plasmid into the cells, re-plating cells at low density 48 h later and selecting stable integrants at 0.2 mg ml⁻¹ zeocin for 10−14 days. RPE1 FRT/TR p53KO cells were co-electroporated, using NEON system (Thermo Fisher Scientific), with flippase-expressing pOG44 plasmid (Thermo Fisher Scientific) and pcDNA5/FRT/TO-neo containing EYFP-tagged PALB2 constructs; stable clones were selected at 0.6 mg ml⁻¹ G418 for 7−10 days. Individual colonies were picked, expanded and tested for protein expression upon induction with doxycycline.

**CRISPR-Cas9-mediated gene knock-outs and endogenous tagging.** Gene knock-outs (KO) were generated following the protocol described in ref. [52]. Briefly, U2OS/TLR and RPE1 p53 null cells were transfected or electroporated with the All-in-One plasmids expressing either EGFP or mCherry fluorescent marker, Cas9D10A nickase and two guide RNAs specific for the gene(s) of interest. Forty-eight hours later, cell cultures were single-cell FACS-sorted for single or dual fluorescence and allowed to form colonies on 96-well plates. To verify the KOs, individual clones were analysed by a combination of high-throughput immuno-fluorescence microscopy, immunoblot and PCR approaches. In case of BRCA1 KO, cell lines, which lacked full-length BRCA1 protein (as detected by immunoblotting) and showed altered pattern of DNA fragments generated in PCR across the targeted genomic locus, were also profiled for their sensitivity to PARP1 inhibitor olaparib.

To tag PALB2 at its endogenous gene locus, RPE1 cells were electroporated with two plasmids: the mCherry-expressing All-in-One Cas9 nickase vector with gRNAs targeting the first exon of PALB2 gene and the vector with the Venus tag-encoding sequence sandwiched between the two homology arms which comprised the sequences flanking the gRNA-targeted site (see Supplementary Fig. 2a for graphic details). In the targeting template, two guanines of the PAM motif for the gRNA-Sense were mutated using side-directed mutagenesis kit (Invitrogen) to avoid nicking of the homology donor. Insertion of the Venus open reading frame at the start of PALB2 coding sequence destroyed the target site for the antisense gRNA. Forty-eight hours after electroporation, the cells that expressed both mCherry and Venus markers (1.3%) were plated on four 96-well plates at a single cell per well density. After 2 weeks, a total of 47 surviving colonies were picked and expanded for PCR and further analyses. According to the PCR analysis, 19 out of 47 clones appeared to have both alleles of PALB2 gene tagged with Venus. PCR fragments were further analysed by restriction digestion and Sanger sequencing. The functionality of endogenously tagged Venus-PALB2 protein was characterised by immunoprecipitation (interaction with BRCA1, BRCA2 and MRG15 proteins) and immunofluorescence (IR-induced focus formation in S/G2 cells).

Sequences of primers used to generate sgRNAs can be found in Supplementary Table 2.

**Antibodies.** Information regarding primary antibodies used in this work can be found in Supplementary Table 3.

**DNA-damage induction and drug treatments.** For ionising radiation treatments, a Faxitron-CellRad (Faxitron Bioptics, LLC) fitted with an aluminium filter for soft X-rays was used. Localised DNA damage was generated by exposure of cells to a UV-A laser beam in laser micro-irradiation experiments performed essentially as described in ref. [53]. In brief, cells were plated on glass-bottomed dishes (Willco-Wells) and pre-treated with 10 µM 5-bromo-2′-deoxyuridine (BrdU, Sigma-Aldrich) for 24−36 h. Laser micro-irradiation was performed using a FluoView 1000 confocal microscope (Olympus) equipped with a 37 °C heating stage (Ibidi) and a 405-nm laser diode (6 mW) focused through a 360 UPlanSApo/1.35 numerical aperture lens. The time of cell exposure to the laser beam was around 250 ms (fast scanning mode). To induce S-phase-specific DSBs, cells were treated with 1 µM camptothecin (Sigma-Aldrich) for 1 h.

**Clonogenic survival assays.** Cells were seeded at low density in six-well plates at two different dilutions, treated with various concentrations of PARPi (olaparib; Stratech Scientific) and allowed to form colonies for 7−10 days. Colonies were stained with 0.5% crystal violet/20% ethanol and counted. Results were normalised to plating efficiencies.

**TLR assays.** To evaluate HR efficiency upon siRNA-mediated depletion of various proteins, the TLR assays based on the system described in ref. [10] were performed. The workflow was as described in ref. [11]. Briefly, U2OS-TLR cells were transfected with control siRNA (targeting firefly luciferase gene) or various other siRNAs and 6−8 h later were co-transfected with expression plasmids containing: infrared fluorescent protein (IFP) and I-SceI endonuclease; and blue fluorescent protein (BFP) together with the donor sequence. Cells were collected ∼72 h after siRNA transfection. Four-colour fluorescent flow cytometry using a BD LSRFortessa cell analyser (BD Biosciences) allowed direct measurements of the percentages of GFP + (HR). A minimum of 10,000 doubly transfected (IFP+ and BFP+) cells were scored for each condition in a minimum of three experiments. Data analysis was performed using FlowJo X (Tree Star).

**Immunoblotting.** Total cell lysates were prepared using SDS lysis buffer (4% SDS, 20% glycerol, and 120 mM Tris-HCl, pH 6.8), heating for 5 min at 95 °C and fragmenting gDNA by passing lysates several times through a 19-gauge needle. Absorbance at 280 nm was measured on NanoDrop (Thermo Fisher Scientific) to determine protein concentration. Samples were resolved on 4−12% Bis-Tris gels NuPAGE gels (Novex, Life Technologies) and transferred onto nitrocellulose membranes (GE Life Sciences) or polyvinylidene difluoride (PVDF). Membranes were blocked with either 1−5% bovine serum albumin (BSA) or 3−5% milk in TBS/0.1% Tween-20 (TBST) and incubated with the relevant primary antibody (Supplementary Table 2) followed by several TBST washes and incubation with corresponding horse radish peroxidase-conjugated secondary antibody (Fisher Scientific). Enhanced chemiluminescence (ECL) reagent (GE Healthcare) was used for detection.

**Immunofluorescence.** For confocal and 3D-SIM microscopy, cells were grown on poly-lysine-coated coverslips; for automated high-throughput IRIF detection and counting using OPERA system, cells were plated in 96-well plates (Cell Carrier, Perkin Elmer). For detection of RPA2 and PALB2, cells were washed with phosphate-buffered saline (PBS), pre-extracted for 2 min on ice in CSK buffer (25 mM HEPES pH 7.4, 50 mM NaCl, 1 mM EDTA, 3 mM MgCl₂, 300 mM sucrose) containing 0.25% Triton X-100, and fixed with 2% formaldehyde for 15 min at room temperature. For staining with all other antibodies, pre-extraction step was omitted. Fixed cells were washed three times with PBS containing 0.1% Tween-20 (PBST), permeabilised with 0.2% Triton X-100 in PBS for 5 min and blocked with 0.2% fish skin gelatin in PBS for 30 min to overnight. One-hour incubation with primary antibodies was followed by three washes with PBST and 30-min incubation with secondary, Alexa-Fluor-conjugated, antibodies (Molecular Probes) at 1:1000 dilution and 1 µg ml⁻¹ containing 4,6 diamidino-2-phenylindole (DAPI, Sigma-Aldrich). After three final washes with PBST, the coverslips were rinsed in deionized water and mounted on glass slides using DAPI-free VectaShield mounting medium (Vector Laboratories).

**Automated high-throughput microscopy for IRIF quantification.** For high-throughput imaging and analyses of focus formation by RPA2, RAD51 and PALB2 proteins, cells were plated on 96-well Cell Carrier (Perkin Elmer) plates at a density of 7500−10,000 cells/well. The following day, cells were either mock-treated or irradiated (6 Gy) and processed for imaging as described under Immunofluorescence method. Imaging was performed on the spinning disk Opera Phenix microscope (Perkin Elmer) using either ×20 or ×40 water immersion objectives. Images were acquired in single optimised focal plane for each of the three channels (405, 488 and 568 nm) in the confocal mode. Image analysis and evaluation was done using Harmony High-Content Imaging and Analysis software. In each experiment, 7−10 fields (>500 cells) per well were acquired and analysed.

**Confocal microscopy.** The confocal images were acquired on an Olympus FV1000 upright confocal microscope using ×40 or ×60 UPlanSApo/1.35 oil objective. For

the experiment presented in Fig. 3b−e, image analysis and quantification were performed using a custom script (https://github.com/gurdon-institute/Intensity_Distribution) for Fiji[54]. This high-throughput macro segments nuclei in the DAPI channel by Otsu thresholding and segments GFP-PALB2 foci in the green (488 nm) and RPA2 foci in red (594 nm) channels by detecting local intensity maxima as seed points to trace uniform regions with a tolerance of half the global channel mean. For each red focus, the distance to the nearest green focus is measured and a call for closeness is made using a cutoff distance of 0.5 μm. Integrated density (area × mean intensity) is measured for each green focus.

**3D/SIM super-resolution microscopy**. Super-resolution 3D SIM imaging of Venus-PALB2 and RPA was performed using a Deltavision OMX 3D SIM System V3 (GE) equipped with 3 EMCCD Cascade cameras from Photometrics, 405, 488, 593 nm diode laser illumination, an Olympus PlanSApo ×100 1.42 NA oil objective, and standard excitation and emission filter sets. Imaging of each channel was performed sequentially using three angles and five phase shifts of the illumination pattern as described in ref. [22]. The refractive index of the immersion oil (Cargille) was adjusted to 1.514 to minimise spherical aberrations. Sections were acquired at 0.125 μm z steps. Raw OMX data were reconstructed and channel registered in the SoftWoRx software (GE). Reconstructions were carried out using channel-specific optical transfer functions (OTFs) and channel-specific K0 angles. OTFs were generated within the SoftWoRx software by imaging 100 nm beads (Life Technologies) using appropriate immersion oils to match the data. Channel registration was carried out using the Image Registration parameters generated within the SoftWoRx software and checked for accuracy by imaging Tetraspeck beads (Life Technologies). Channel registration was accurate to within one pixel. Further data analysis was performed using a Fiji plugin Surrounding Blobs (https://github.com/gurdon-institute/Surrounding_Blobs) that we developed for high-throughput segmentation and measurement of 3D foci in OMX images. Local intensity maxima are mapped in 3D to seed assignment of nearby thresholded voxels, and the number of green (488 nm) foci within 200 nm of each red (593 nm) focus is measured.

**6xHis-MBP-ChAM purification**. 6xHis-MBP and 6xHis-MBP-ChAM protein variants were induced in BL-21 DE3 RIL cells using 200 μM IPTG overnight at 16 °C. Cell pellets were resuspended in lysis buffer (25 mM phosphate buffer pH 7.4, 300 mM NaCl, 0.1% Triton (v/v), 10% glycerol (v/v), 5 mM β-mercaptoethanol, 1× Protease Inhibitor mix [284 ng ml$^{-1}$ leupeptin, 1.37 μg ml$^{-1}$ pepstatin A, 170 μg ml$^{-1}$ phenylmethylsulfonyl fluoride and 330 μg ml$^{-1}$ benzamindine], 1 mM AEBSF and 5 μg ml$^{-1}$ DNaseI) Cells were lysed by sonication and lysozyme treatment. Clarified lysate was applied to a HiTrap chelating column (GE Healthcare) pre-loaded with Ni$^{2+}$ ions, or Ni-NTA resin (Qiagen). After extensive washing 6xHis-MBP-ChAM was eluted using 300 mM imidazole and concentrated a 30 K MWCO centrifugation device (Amicon). 6xHis-MBP-ChAM was further purified on by size exclusion chromatography using a superdex 200 Increase 10/300 in SEC buffer (20 mM Tris pH 7.5, 150 mM NaCl, 1 mM dithiothreitol (DTT), 5% glycerol) and the main mono-disperse protein containing peak was collected, concentrated, flash frozen in liquid nitrogen and stored at −80 °C.

Protein concentrations were determined via absorbance at 280 nm using a Nanodrop 8000 (Thermo Scientific), followed by SDS-PAGE and InstantBlue (Expedeon) staining with comparison to known amounts of control proteins.

**Chemical ubiquitylation of histone H2A**. Mutant human Histone H2A engineered with a single cross-linkable cysteine (H2A K15C) was chemically ubiquitylated essentially as described[25,55]. Briefly, an alkylation reaction was assembled with H2A K15C (700 μM), 6xHis-TEV-ubiquitin G76C (700 μM) and 1,3-dibromoacetone (4.2 mM, Santa Cruz) in 250 mM Tris-Cl pH 8.6, 8 M urea and 5 mM TCEP (tris(2-carboxyethyl)phosphine) and allowed to react for 16 h on ice. The reaction was quenched by the addition of 10 mM β-mercaptoethanol and pH adjusted to 7.5. Chemically ubiquitylated H2A (H2A Kc15ub) was purified using a HiTrap SP HP column (GE Healthcare) and 6xHis-TEV-H2AKc15ub containing fractions were pooled and enriched over a HiTrap chelating column (GE Healthcare) pre-loaded with Ni$^{2+}$ ions. The 6xHis tag was removed by TEV cleavage and subsequent Ni$^{2+}$ column subtraction. The resulting flow-through was dialysed against a 2 mM β-mercaptoethanol/dH20 solution and lyophilised. H2A Kc15ub was refolded and wrapped into NCPs as described below.

**NCP reconstitution**. Nucleosome core particles (NCPs) were reconstituted essentially as described[25]. Briefly the four core human histones were expressed in BL-21 DE3 RIL cells and purified from inclusion bodies as described[56]. 145 bp Widom-601 DNA fragments for wrapping nucleosomes were generated by either cleavage of pUC57-8 × 145 bp Widom-601 DNA[57] and purification[56] or by PCR-based amplification[58,59]. Briefly for PCR amplification, 384 100-μl PCR reactions using Pfu polymerase and HPLC pure oligos (IDT) were pooled, filtered and purified using a ResourceQ column and DNA was eluted using a salt gradient.

For octamer formation, histones were mixed at equimolar ratios in unfolding buffer (7 M Guanidine HCl, 20 mM Tris pH 7.5, 5 mM DTT) prior to dialysis to promote refolding into 2 M NaCl, 15 mM Tris pH 7.5, 1 mM EDTA, 5 mM β-mercaptoethanol. Octamers were selected by gel filtration chromatography and assembled into NCPs via salt gradient dialysis[60]. Soluble NCPs were partially polyethylene glycol (PEG) precipitated[25,61] and resuspended in 10 mM HEPES pH 7.5, 100 mM NaCl, 1 mM EDTA, 1 mM DTT. NCP formation and quality was checked by native gel electrophoresis and used within 1 month of wrapping.

**NCP pull-down assays**. Pull-down assays were either performed with PALB2 immobilised directly from expressed *Escherichia coli* extract on Ni-NTA resin (Fig. 4f, g and Supplementary Fig. 5d) or with fully purified 6xHis-MBP-tagged PALB2 ChAM immobilised after gel filtration chromatography, concentration, quantification and freezing (Fig. 4a and Supplementary Fig. 5a, b). For Fig. 4f, g and Supplementary Fig. 5d, 6xHis-MBP ChAM variants were lysed as for 6xHis-MBP-ChAM purification and lysate bound in batch to Ni-NTA beads to saturation (roughly 12 μg protein per μl beads). Beads were washed extensively and amount of protein quantified and normalised by comparison on SDS-PAGE, against known purified MBP-ChAM. Pre-immobilised beads were used immediately or the next day, diluted so that 35−40 μg of MBP-ChAM protein per 15 μl of Ni-NTA beads was used per pull-down.

For purified his-MBP-ChaM pull-downs, 25 μg of 6xHis-MBP-tagged PALB2 ChAM proteins were immobilised on Ni-NTA resin (Qiagen) or Amylose resin (NEB) for 1 h. All pull-down assays were incubated with 2.2 μg of NCP complex in pull-down buffer (20 mM Tris-Cl pH 7.5, 120 mM KCl, 2 mM β-mercaptoethanol, 10% glycerol (v/v), 0.01% NP-40 (v/v), 100 μg/ml BSA, 15 mM imidazole) for 2 h at 4 °C. Pull-downs were washed thoroughly in pull-down buffer and resuspended directly in 2× SDS loading buffer. All pull-down assays were repeated at least twice, with a single immunoblot displayed.

**Statistics and reproducibility**. Statistical analyses were performed using Graph-Pad Prism version 8 for Mac OS X, GraphPad Software (La Jolla, CA, USA).

**Reporting summary**. Further information on research design is available in the Nature Research Reporting Summary linked to this article.

## Data availability

All the data supporting the findings of this study are either available within the paper and its Supplementary Information files or can be obtained from the authors upon request.

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

## Acknowledgements

The authors are grateful to Bin Xia for a kind gift of anti-PALB2 antibody, Ting-Wei Will Chiang for generating Cas9[D10A] All-in-One vector for *BRCA1* knock-out, and Qian Wu for cloning and initial optimisation of the expression and purification conditions of the recombinant ChAM domain. We thank Josep Forment and Israel Salguero Corbacho for many discussions and critical reading of the manuscript, Kate Dry for editorial help and all S.P.J. laboratory members for support and advice. We are very grateful to Alessandro Costa (Crick Research Institute) for supporting M.D.W. during the initiation of the project. The S.P.J. lab is funded by the Cancer Research UK (CRUK) Program Grant C6/A18796, and Wellcome Investigator Award 206388/Z/17/Z. Core infrastructure funding was provided by CRUK Grant C6946/A24843 and Wellcome Grant WT203144. S.P.J. receives salary from the University of Cambridge. R.B. is funded by CRUK Program Grant C6/A18796, E.R.-G. is funded by Action for A-T and the A-T Children's Project. M.D.W.'s work is supported by Wellcome [210493] and the University of Edinburgh. The Wellcome Centre for Cell Biology is supported by core funding from Wellcome [203149] and Multi-User Equipment grant [101527] for the Edinburgh Protein Production Facility.

## Author contributions

R. Belotserkovskaya conceived the project and the studies comprising it with advice from S.P.J. R. Belotserkovskaya also planned, performed and analysed the cell culture-based experiments. M.D.W. generated recombinant nucleosome core particles and MBP-ChAM, and carried out in vitro pull-down experiments. G.M. purified proteins for

in vitro assays. E.R.G. helped with plasmid amplification and cell line maintenance, and performed one of the clonogenic survival assays. N.L. provided extensive assistance with confocal and super-resolution microscopy, especially image acquisition and processing of the 3D-SIM-acquired images. R. Butler wrote custom scripts for image analyses in Fiji. R. Belotserkovskaya, M.D.W. and S.P.J. wrote the manuscript.

## Competing interests

S.P.J. declares that he is founder and shareholder of Mission Therapeutics Ltd, Adrestia Therapeutics Ltd, Ahren Innovation Capital LLP, and Carrick Therapeutics Ltd. The other authors declare no competing interests.
