## [Peer Review File · Nature Communications]

Reviewers' comments:

Reviewer #1 (Remarks to the Author):

In this paper, Belotserkovskaya et al. propose that the profound effects of BRCA1 loss on HR are mainly due to defective recruitment of the PALB2-BRCA2 RAD51 complex. According to this model, 53BP1 occludes PALB2 recruitment to resected DNA by binding avidly to the nucleosome acidic patch. This prevents PALB2, via its ChAM domain, from direct association with the H2A-H2B acidic patch at sites of DSBs. Their finding that in BRCA1/53BP1 deficient (but not in WT cells), PALB2 recruitment to IR induced foci becomes dependent on the ChAM domain of PALB2 is interesting. However, the model they propose in which the 53BP1-acid patch interaction occludes the PALB2-ChAM is overly simplistic and does not seem consistent with previous publications. For example, loss of Shieldin, which acts downstream of 53BP1, also rescues HR in BRCA1 deficient cells. In Shieldin KO cells, 53BP1 is still avidly bound to chromatin. Why isn't PALB2 blocked by 53BP1 in this case? Similarly loss of ATM dependent 53BP1 phosphorylation doesn't alter 53BP1 binding to chromatin, but presumably enables the alternative mode of PALB2 recruitment. They also argue that similar to 53BP1 depletion, loss of RNF169 enhances PALB2 IRIF. However, loss of RNF169 also increases 53BP1 foci, which should occlude PALB2 according to their model. Finally, RNF168 mediated H2A15ub, which is reliant on RNF168's interaction with the acidic patch, does not block but is necessary for PALB2 recruitment. In summary, there are many factors whose interaction with the acidic patch does not prevent, but actually promote PALB2 recruitment to DSBs in BRCA1 deficient cells. Thus, the "occlusion model" must be significantly strengthened.

Specific comments:

1. The authors propose that several basic residues within the ChAM domain of PALB2 mediate binding to the nucleosome acidic patch. They show that mutating these residues reduced the pulldown of reconstituted nucleosomes. While it may be that this was due to altered electrostatic interactions between ChAM and the acidic patch, an alternative explanation is that mutating these residues changes the overall structure of PALB2, precluding binding to nucleosomes. The authors need to show that mutating conserved but non-charged residues within the ChAM domain do not affect its interaction with acidic patch before concluding that binding between ChAM and nucleosomes is based on electrostatic interactions.

2. In response to DNA damage, H2A is ubiquitylated by RNF168 in a manner dependent on its interaction with the acidic patch, which subsequently promotes 53BP1 binding. While it is possible that an isolated ChAM domain can bind unmodified histones, it is equally possible that binding is influenced by histone PTMs. The authors should test whether ubiquitylation of H2A has any impact on the engagement between ChAM and acidic patch.

3. The authors speculate that 53BP1 blocks PALB2 from binding to chromatin. However, it has been shown previously by others that a 53BP1 mutant lacking all N-terminal phosphorylation sites is functionally equivalent to 53BP1-null. This would suggest that 53BP1 itself cannot block PALB2 binding to chromatin.

4. What is the significance of R414 mutation in breast cancer patients? As the authors suggested, mutating the ChAM domain has no impact on PALB2 response in BRCA1-proficient cells so if the tumor is BRCA1 WT, then it is presumably harmless. If the tumor is BRCA1 mutant, on the other hand, the R414 mutation would be detrimental, so why would it be selected?

5. RNF169 has been demonstrated by several groups to promote HR by virtue of its ability to displace 53BP1. Conversely, 53BP1 foci formation is increased by RNF169 depletion. Thus, it is unexpected that deletion of RNF169 would increase PALB2 binding. Moreover, the authors clearly showed in fig.5c that in BRCA1/53BP1 cells, depleting RNF169 had no effect on PALB2 foci. If so, how can it be that RNF169 competes with PALB2 for chromatin binding?

6. The authors referred to Luijsterburg et al, stating that H2AK15Ub is not important for PALB2 to recognize nucleosomes. However, the authors actually showed that purified PALB2 cannot bind K63-linked Ub chains but binding to Ub-modified NCPs was not tested. Given that 53BP1 efficiently recognizes ubiquitylated histones only in the context of NCP, it is unclear at the moment whether PALB2 can or cannot bind nucleosomes in a ubiquitylation-dependent manner. In fact, the Luijsterburg et al clearly stated that "PALB2 recruitment requires H2A ubiquitylation at K13/K15", although they propose a more indirect mechanism through RNF168-mediated bridging between PALB2 and chromatin. The authors should show convincing data supporting the notion that ubiquitylation is not important for PALB2 binding to chromatin.

7. The authors published earlier that deletion of the 53BP1 effector complex Shieldin rescues BRCA1 mutant cells by de-repressing resection. If that is correct, and there is a lot of convincing evidence to suggest that it is so, then how can it be that BRCA1 deficient cells show no measurable defect in end resection? Does deletion of Shieldin components in BRCA1-deficient cells allow for PALB2 recruitment via its ChAM domain. If so, why is 53BP1 chromatin binding not occluding this recruitment.

Reviewer #2 (Remarks to the Author):

Loss of BRCA1 impairs HR and sensitizes cells to PARP inhibitor olaparib, a chemotherapeutic drug that has been approved to treat BRCA1/2-deficient ovarian, breast and other cancers. It has been reported that loss of 53BP1 can promote resistance of BRCA1-deficient cells to PARPi. In this manuscript, the authors investigated the mechanisms underlying this resistance. They reported that loss of 53BP1 partially rescues HR in BRCA1- but not PALB2- or BRCA2-deficient cells. They showed that depletion of 53BP1 rescues PALB2 focus formation in BRCA1-deficient cells and that this rescue is dependent upon the chromatin association motif (ChAM) of PALB2. They went on to further show that ChAM of PALB2 interacts with the acidic patch of H2A-H2B. They concluded that 53BP1 and RNF169 compete with PALB2 binding to chromatin and that loss of 53BP1 or RNF169 in BRCA1-deficient cells allows PALB2 better access to bind the surfaces (acidic patch) on nucleosomes. This rescue in PALB2 binding to chromatin contributes to the resistance of BRCA1- and 53BP1-depleted cells to PARPi.

There are a number of major issues (below) with this manuscript and as a result, it is premature to be considered for publication in Nature Communications.

Major issues:

1. The concept that ChAM of PALB2 binds nucleosomes is not new. The authors claim that ChAM of PALB2 binds the acidic patch of H2A-H2B, however the evidence supporting this claim is weak and insufficient. The authors need to show if there is a direct and physical interaction between ChAM and the acidic patch. The authors also need to show if such an interaction also exists with full-length PALB2.

2. The quality of the data presented in Fig. 4a is very poor. The recombinant MBP-ChAM protein seems to be degraded. Why is there a huge variation in the amount of H2B, H4 and Δ H4 in the input? The lack of Δ H4 NT in MPB-ChAM pull-down (last lane) seems to suggest that the N-terminal tail of H4 may be engaged in binding with ChAM, which would not support their claim that ChAM interacts with the acidic patch of H2A-H2B.

3. In Fig. 4d, the authors showed that ChAM containing a single mutation of R414A or R421A is as defective as ChAM-4M in binding nucleosomes. The authors suggest that ChAM binds the acidic patch of H2A-H2B through electrostatic interaction. If so, does a conservative substitution of

R414K or R421K affect ChAM interaction with nucleosomes? Does a conservative substitution of R414K or R421K affect the ability of PALB2 to confer sensitivity to Olaparib, HR activity and RAD51 foci, etc?

4. In Fig. 3, the authors showed that PALB2 Δ ChAM does not confer olaparib sensitivity in the presence of BRCA1 but does so in the absence of BRCA1 and 53BP1. Previously it has been reported that PALB2 Δ ChAM does confer olaparib sensitivity in cells proficient for BRCA1 (Bleuyard et al. 2011). Although the authors made no mention of this discrepancy in the manuscript, they need to address it. Is their observation on PALB2 Δ ChAM and olaparib sensitivity unique to RPE-p53-KO cells? In addition, what is the effect of PALB2 Δ ChAM on HR activity as well as BRCA2 and RAD51 foci in the presence or absence of BRCA1 and 53BP1?

5. In Fig. 2 and S2f, RPE1 Venus-PALB2 clone #1 was used in their analysis. However, Fig. S2b clearly showed that the clone #1 does not contain the expected 3.5-kb band that indicates Venus insertion. It is puzzling and concerning how the authors were able to show that this clone #1 behaves the same as the clone #15 that does have the Venus insertion.

6. The authors claim that both 53BP1 and RNF169 are able to compete with PALB2 binding to chromatin. It has been reported that RNF169 limits 53BP1 deposition to DSBs although it does not completely abrogate 53BP1 at DSBs (An et al. 2018). Therefore one might expect PALB2 focus count to be higher in BRCA1/53BP1-deficient cells depleted for RNF169 compared to BRCA1/53BP1-deficient cells expressing siControl. Instead the authors showed that depletion of RNF169 did not affect PALB2 focus count in BRCA1/53BP1-deficient cells (Fig. 5c), which does not appear to support their claim.

7. Based on their finding that depletion of RNF169 leads to an increase in PALB2 focus formation in WT cells, the authors suggest that RNF169 and PALB2 compete for their binding to the acidic patch of H2A-H2B. This conclusion is poorly supported and needs to be further substantiated. For example, RNF169 binds to the acidic patch through its LR-motif. Do mutations in LR-motif that abolish RNF169 binding to the acidic patch lead to an increase in PALB2 focus formation?

Other issues:

1. Lack of explanation of asterisks in many figures.
2. Figure legends need to be better described. For example, in Fig. 3c and 3e, what is the meaning of the number after WT, Δ ChAM, Δ MRG and Δ Ch/M?
3. Which of PALB2 Δ ChAM, Δ ChAM13 or Δ ChAM13, is used in clonogenic survival assays in Fig. 3f, 3g, 4e-4g?
4. Fig. 5d, the status of BRCA1 is not correctly indicated.

Reviewer #3 (Remarks to the Author):

In this paper by Jackson and colleagues, the authors investigate the molecular mechanism behind the suppression of BRCA1 deficient cells hypersensitivity to Olaparib by 53BP1 inactivation. They reasoned that, due to the almost normal resection observed in RPE1 cells defective in BRCA1, the suppression may be due to later events during recombination such as Rad51 loading. Indeed, they can see a strong defect in HR and Rad51 loading on those cells, that is readily suppressed by 53BP1 inactivation. Interestingly, such suppression is not observed in PALB2 or BRCA2 defective cells. Hence, the authors set to study if in the absence of BRCA1 PALB2 accumulation at DSBs is compromised and, moreover, if this is alleviated by 53BP1 depletion. Indeed, this is the case, prompting them to propose that in the absence of BRCA1 and 53BP1 PALB2 loading is achieved by alternative means. Biochemical studies support this idea, indicating that PALB2 can bind the acidic patch of histones. Indeed, this binding seems to be mostly dependent on PALB2 ChAM domain.

Overall, the data are well presented, the results clear and the model integrate all the data. My main concern is if the data as they are can really completely exclude an effect of 53BP1 on alleviating the mild defect in DNA resection known to be associated with BRCA1 deficiency. Even if such defect is too small to be visualized by RPA foci or FACs it cannot be excluded that it can contribute to the Rad51/PALB2 recruitment. Alternatives ways to analyze this will be to use a method that quantify resection processivity or to repeat some critical experiments with KO of RIF1, REV7 or Shieldins. As the whole shieldin complex is required to inhibit resection, any effect on suppression of the BRCA1 resection defect should be equally observed in any of those KOs. However, on those KOs cells, 53BP1 will still be attached to chromatin, likely blocking PALB2 binding through the ChAM domain. Thus, in this setup, a separation of function on resection/PALB2 recruitment should be observed.

In any case, even if 53BP1 alleviates BRCA1 sensitivity to olaparib through both resection- and PALB2- dependent mechanisms, this does not diminish the main message, that in the absence of 53BP1 the ChAM domain of PALB2 can now facilitates its recruitment to DNA breaks.

Minor comments:

On Page 5, the laser line experiment is mentioned a couple of times as S3b, and it should be referred as S3c

Page 5, end of second paragraph, a period is missing between "(Fig. 2c; supplementary movies)" and "Together"

Reviewer #4 (Remarks to the Author):

This is an original and provocative study that gives more insight in the regulation of DNA DSB repair in 53BP1;BRCA1 double deficient cells. Similar to the recently published work by Zong et al. (reference 37), the authors investigated how loss of 53BP1 can restore RAD51 loading in BRCA1 deficient cells. They show that PALB2 recruitment to RPA-IRIF positive cells is increased in 53BP1;BRCA1 double deficient cells. In contrast to Zong et al, they claim that restoration of DNA end-resection does not play a major role in this process. This is a bold claim that requires more experimental evidence. The authors do not convincingly prove that the restoration of PALB2 recruitment is independent on end-resection. In addition, is not clear how 53BP1-loss-mediated increased PALB2 binding to the H2A-H2B acidic patch can lead to increased focal accumulation of PALB2 at sites of DNA damage. Nevertheless, the data in this study do suggests that 53BP1-loss-mediated restoration of HR in BRCA1 deficient cells goes beyond increased end-resection. This adds to our insight in the regulation of DNA double strand break repair, a process that is important for tumor suppression and treatment response.

Specific comments

1. Based on the data presented I am not convinced that BRCA1 `has a much weaker impact on resection than on RAD51 filament formation ..'. Data from several groups have confirmed the end-resection defects in BRCA1 deficient cells shown in reference 10 (Bunting 2010), including Drost et al. (2011 PMID: 22172724, 2016 PMID: 27454287) and He et al. (2018, PMID: 30464262). Discrepancies with the data in this manuscript and other findings may stem from the use of different experimental settings or techniques. E.g. measuring nuclear RPA intensity (Fig S1F) may not be a very sensitive approach. There should be stronger evidence that increased PALB2 recruitment to damaged DNA is not due to increased end-resection and by itself sufficient to restore HR (e.g. with the help of something like the PALB2-RNF8FHA fusion protein from reference 37). If the authors cannot provide more convincing experimental evidence, their conclusions should be adjusted. Independent of possible additional data, they should discuss their findings in relation to previous observations that do suggest a role for restoration of end-resection.
2. It is not clear to me how increased binding of PALB2 to the H2A-H2B acid patch can increase

focal accumulation at sites of DNA damage. The authors speculate that may be mediated by RNF168. PALB2 interacts with RNF168 via its WD40 domain, not via the Cham domain (Luijsterburg et al., PMID: 28240985). Does PALB2 Cham domain mutation then indirectly decrease the interaction with RNF168? Or does 53BP1 loss (also?) directly affect RNF168? The authors should at least discuss their findings in relation to the results from Luijsterburg et al.

Minor comments

1. In figure 1 the authors show that 53BP1 depletion does not rescue the HR defect of PALB2 or BRCA2 depleted cells. The latter is in line with the fact that loss of 53BP1 does not rescue proliferation of Brca2 deficient MEFs (reference 11, PMID: 20453858), which should be acknowledged.
2. It is not clear to me why it is emphasized on page 4 that 53BP1 inactivation rescues RAD51 IRIF formation in BRCA1 deficient RPE1 cells. This is a well know feature of 53BP1;BRCA1 double deficient cells (described in many papers, including the references 10 and 11) and also known for RPE1 (Durocher lab). If this is merely to validate their experimental system, this should be acknowledged.
3. For clarity it would be good to indicate in figures 3 and 4 (and not only in the legends) which graphs belong to 53BP1;BRCA1 proficient and deficient cells.

We thank the reviewers for their incisive comments and for raising a number of important points related to our work and thus helping us to improve the quality of the manuscript by addressing these issues.

Reviewer #1

In this paper, Belotserkovskaya et al. propose that the profound effects of BRCA1 loss on HR are mainly due to defective recruitment of the PALB2-BRCA2 RAD51 complex. According to this model, 53BP1 occludes PALB2 recruitment to resected DNA by binding avidly to the nucleosome acidic patch. This prevents PALB2, via its ChAM domain, from direct association with the H2A-H2B acidic patch at sites of DSBs. Their finding that in BRCA1/53BP1 deficient (but not in WT cells), PALB2 recruitment to IR induced foci becomes dependent on the ChAM domain of PALB2 is interesting. However, the model they propose in which the 53BP1-acid patch interaction occludes the PALB2-ChAM is overly simplistic and does not seem consistent with previous publications. For example, loss of Shieldin, which acts downstream of 53BP1, also rescues HR in BRCA1 deficient cells. In Shieldin KO cells, 53BP1 is still avidly bound to chromatin. Why isn't PALB2 blocked by 53BP1 in this case? Similarly, loss of ATM dependent 53BP1 phosphorylation doesn't alter 53BP1 binding to chromatin, but presumably enables the alternative mode of PALB2 recruitment. They also argue that similar to 53BP1 depletion, loss of RNF169 enhances PALB2 IRIF. However, loss of RNF169 also increases 53BP1 foci, which should occlude PALB2 according to their model. Finally, RNF168 mediated H2A15ub, which is reliant on RNF168's interaction with the acidic patch, does not block but is necessary for PALB2 recruitment. In summary, there are many factors whose interaction with the acidic patch does not prevent, but actually promote PALB2 recruitment to DSBs in BRCA1 deficient cells. Thus, the "occlusion model" must be significantly strengthened.

We thank Reviewer 1 for his/her comments and suggestions, which we address in detail below.

Specific comments:

1. The authors propose that several basic residues within the ChAM domain of PALB2 mediates binding to the nucleosome acidic patch. They show that mutating these residues reduced the pulldown of reconstituted nucleosomes. While it may be that this was due to altered electrostatic interactions between ChAM and the acidic patch, an alternative explanation is that mutating these residues changes the overall structure of PALB2, precluding binding to nucleosomes. The authors need to show that mutating conserved but non-charged residues within the ChAM domain do not affect its interaction with acidic patch before concluding that binding between ChAM and nucleosomes is based on electrostatic interactions.

We apologise for the wording of the original manuscript. We did not intend to assert that the interaction was only mediated by electrostatic interactions, and have now amended the text in our revised manuscript (see the new section entitled “PALB2 ChAM domain contacts the nucleosome acidic patch” on pp.8-9 and p.11 for discussion). To address the issue raised by the reviewer, we performed an unbiased alanine scanning experiment to identify the key regions required for nucleosome-binding by the ChAM region. We now confirm that several residues are important for this interaction including non-charged residues (see new fig. 4h and fig S5c). We find that these residues cluster around R414, which may be an acidic patch-binding arginine anchor, and we discuss these new findings in the text. We now conclude that binding involves charge-interactions but is also dependent on other, likely structural features of the region to directly recognise the nucleosome acidic patch. This is consistent with strong evolutionary sequence conservation in the region. We are further investigating this interaction through structural approaches, but we believe this is beyond the scope of this manuscript.

Notably, our pulldown experiments using recombinant purified NCP and ChAM variants correlate very well with recently published chromatin fractionation studies (Bleuyard et al, 2018), in which it is demonstrated that cancer associated mutations in the same region, result in reduced association of PALB2 with chromatin. Our mutagenesis throughout the ChAM domain is unlikely to markedly affect the folding of the domain: upon purification, all the basic residue mutant proteins are as soluble and mono-disperse as the wild-type variant (for example see below size exclusion chromatography traces; main peak at ~16ml pooled for all experiments).

2. In response to DNA damage, H2A is ubiquitylated by RNF168 in a manner dependent on its interaction with the acidic patch, which subsequently promotes 53BP1 binding. While it is possible that an isolated ChAM domain can bind unmodified histones, it is equally possible that binding is influenced by histone PTMs. The authors should test whether ubiquitylation of H2A has any impact on the engagement between ChAM and acidic patch.

and 6. The authors referred to Luijsterburg et al, stating that H2AK15Ub is not important for PALB2 to recognize nucleosomes. However, the authors actually showed that purified PALB2 cannot bind K63-linked Ub chains but binding to Ub-modified NCPs was not tested. Given that 53BP1 efficiently recognizes ubiquitylated histones only in the context of NCP, it is unclear at the moment whether PALB2 can or cannot bind nucleosomes in ubiquitination-dependent manner. In fact, the Luijsterburg et al clearly stated that “PALB2 recruitment requires H2A ubiquitylation atK13/K15”, although they propose a more indirect mechanism through RNF168-mediated bridging between PALB2 and chromatin. The authors should show convincing data supporting the notion that ubiquitylation is not important for PALB2 binding to chromatin.

We thank the reviewer for raising these issues. As requested, we have performed the requested experiment (fig. S5a) and have observed that the ChAM domain interacts equally well with both unmodified and H2AK15ub-containing nucleosomes. As we discuss in our revised text (p.8), these data support a model in which RNF168 mediated histone H2A ubiquitylation does not enrich or exclude the ChAM domain of PALB2 directly, and RNF168 mediated recruitment of PALB2 is likely via a direct

interaction between PALB2 and RNF168, as suggested by Luijsterburg et al (2017). We do not exclude the possibility that PALB2 binding may be influenced by histone PTMs, but in our in vitro system the ablation of the acidic patch in the NCP is sufficient to reduce its interaction with the PALB2 ChAM.

3. The authors speculate that 53BP1 blocks PALB2 from binding to chromatin. However, it has been shown previously by others that a 53BP1 mutant lacking all N-terminal phosphorylation sites is functionally equivalent to 53BP1-null. This would suggest that 53BP1 itself cannot block PALB2 binding to chromatin.

Based on the finding that depletion of 53BP1 in BRCA1-deficient cells restores PALB2 focus formation to nearly control levels, while in BRCA1-proficient cells it results in strong enhancement of focus formation by PALB2 in S/G2 cells, we hypothesised that 53BP1 might physically interfere with PALB2 accumulation at sites of DNA damage, with this blocking by 53BP1 being especially potent when BRCA1 is not present to counteract 53BP1 binding to chromatin in damaged S/G2 cells. The 53BP1 mutant lacking N-terminal phosphorylation sites will be defective at recruiting RIF1 and other components of the Shieldin axis to DSBs. Notably, previous studies have shown that RIF1 loss does not precisely mimic the loss of 53BP1 in that its loss rescues BRCA1 deficiency less strongly than 53BP1 loss does (Feng et al., 2013; Escibano-Diaz et al., 2013). Please also see our response to the point 7, where we describe results of systematic co-depletion of BRCA1 and 53BP1-Shieldin axis components. Based on these experiments, we conclude that 53BP1 binding by itself to DSB-adjacent chromatin is sufficient to interfere with efficient PALB2 focus formation.

4. What is the significance of R414 mutation in breast cancer patients? As the authors suggested, mutating the ChAM domain has no impact on PALB2 response in BRCA1-proficient cells so if the tumor is BRCA1 WT, then it is presumably harmless. If the tumor is BRCA1 mutant, on the other hand, the R414 mutation would be detrimental, so why would it be selected?

We thank the reviewer for this question. At present, there are insufficient data available to allow one to make clear conclusions on the impact of PALB2 R414

mutations in cancer patients – that is, whether these are passenger or driver mutations. We could find one report mentioning familial breast cancer with R414X mutation as well as R414Q mutation as a variant of unknown significance (Hellebrand et al., 2011). It is conceivable that the R414Q mutation reduces PALB2 interaction with chromatin, thereby reducing the efficiency of HR and increasing chromosome translocations and genome instability and thus contributing to malignant transformation, both in BRCA1-proficient and deficient genetic backgrounds.

5. RNF169 has been demonstrated by several groups to promote HR by virtue of its ability to displace 53BP1. Conversely, 53BP1 foci formation is increased by RNF169 depletion. Thus, it is unexpected that deletion of RNF169 would increase PALB2 binding. Moreover, the authors clearly showed in fig.5c that in BRCA1/53BP1 cells, depleting RNF169 had no effect on PALB2 foci. If so, how can it be that RNF169 competes with PALB2 for chromatin binding?

It is indeed the case that the first studies aiming at understanding the function of RNF169 in the DDR established that RNF169, as well as other proteins containing LR motifs and UBDs (including RAP80 and RAD18) can be recruited in a ubiquitin dependent manner to DSBs (Panier et al., 2012; Chen et al., 2012; Poulsen et al., 2012). *In vitro* studies that followed, demonstrated that RNF169 could bind ubiquitylated NCPs directly with the highest affinity (Kitevski-LeBlanc et al., 2017; Hu et al., Mol Cell 2017). Based on several observations: 1, depletion of RNF169 resulted in elevated 53BP1 focus counts; 2, RNF169 overexpression attenuated 53BP1 focus formation; and 3, depletion of RNF169 dampened HR efficiency, it was concluded that RNF169 might promote HR. However, detailed analysis of the effects of RNF169 on the DSB repair (An et al., 2019) has revealed that the relationship between RNF169 and 53BP1 appears to be far more complicated than originally postulated. Overexpression of RNF169 leads to over-resection, increased RAD52 loading and enhanced mutagenic DSB repair by single-strand annealing (SSA). Thus, An et al. (2019) proposed the role for RNF169 as a molecular rheostat that, together with other factors, such as 53BP1 and RAP80, controls the extent of DSB processing and repair pathway choice. Importantly, this study also showed that overexpression of RNF169 in HR deficient cells (depleted for BRCA1, PALB2 or

BRCA2) stimulates SSA repair (An et al., 2019). Moreover, it has been reported that loss of PALB2 and BRCA2 leads to a strong increase in SSA, suggesting that SSA is normally suppressed by HR (Tutt et al., 2001; Obermeier et al., 2016). Therefore, our finding that RNF169 depletion can enhance PALB2 focus formation is in line with previously published data.

In our original data, we found that while we consistently observed that RNF169 depletion had much stronger effects on increasing PALB2 foci in BRCA1/53BP1-proficient (fig. 5a-b) than in BRCA1/53BP1 KO (fig. 5c-d) cells, in the latter background, these effects were not statistically significant. However, upon repeating the experiment several more times to accumulate further data, we now establish statistically significant differences between the samples, as now reflected in our revised fig. 5c-d. Moreover, we tested if RNF169 could directly interfere with PALB2 focal recruitment by overexpressing RFP-RNF169, full-length (FL) or Δ LRM (mutant lacking the C-terminal chromatin binding LR motif (Panier et al., 2012)), in either WT or BRCA1/53BP1 KO cells and quantifying PALB2 IRIF. The results are in new fig. 5e-g. In brief, overexpression of RNF169 FL had an inhibitory effect on PALB2 focus formation while RNF169 Δ LRM did not inhibit PALB2 IRIF formation. Altogether, our data support the notion of RNF169 competing with PALB2 for chromatin binding.

7. The authors published earlier that deletion of the 53BP1 effector complex Shieldin rescues BRCA1 mutant cells by de-repressing resection. If that is correct, and there is a lot of convincing evidence to suggest that it is so, then how can it be that BRCA1 deficient cells show no measurable defect in end resection? Does deletion of Shieldin components in BRCA1-deficient cells allow for PALB2 recruitment via its Cham domain. If so, why is 53BP1 chromatin binding not occluding this recruitment.

We would like to emphasize that in the study the reviewer is referring to by Dev et al. (2018), we proposed that the role of Shieldin is not only in limiting BRCA1-mediated end processing but also in interfering with loading of BRCA2 and RAD51 onto the resected DNA and thus counteracting HR (Dev et al., 2018). If our explanation of the results in the original manuscript was not clear enough, we do apologise; we have now amended the text to try to make it clearer (last part of the discussion, pp.11-12).

In the text of our original manuscript, we cited previously published papers in which it was clearly established that BRCA1 controls the kinetics of DNA end-resection but is not absolutely required for the actual process. In two of those studies, the extent of resection in BRCA1-depleted cells was evaluated using the technique known as single molecule analysis of resection tracks (SMART), which allows accurate measurement of the physical length of ssDNA (Cruz-Garcia et al., 2014; Densham et al., 2016). Even though the tracks are shorter in BRCA1-depleted cells, their lengths are still measured in tens of microns encompassing tens of thousands of nucleotides, which would accommodate thousands of RPA complexes. Making use of this information, we rationalise why we detect RPA signals by the indirect immunofluorescence approach in our experiments. Moreover, we image cells at 5-8 hours after irradiation, which would be a relatively late time frame for accurate evaluation of the efficiency of DNA end-processing. At these later time points, we consistently observe minor differences in RPA signal intensity or focus counts across all samples. Our data are in good agreement with the results presented in fig. 2D of Polato et al. (2014), where DNA end-resection was tested by measuring BrdU signals following irradiation of BRCA1-deficient MEFs. Even though we do not see striking defect in RPA loading in BRCA1-deficient at the late time points, we consistently find defective RAD51 focus formation in those cells.

In regard to the effect of Shieldin depletion on PALB2 focus formation in BRCA1-depleted cells, we have performed experiments whereby we co-depleted BRCA1 with various components of 53BP1-Shieldin complex. To evaluate PALB2 focus formation, these experiments had to be done via siRNA-mediated depletions of 53BP1, RIF1, MAD2L2/REV7 and Shieldin components in cells with endogenously Venus-tagged PALB2. The results of these experiments are presented in new fig. S8. In this set up, we observed that the effect of rescuing PALB2 foci in BRCA1-depleted cells became gradually “diluted” as experiments progressed from 53BP1 to Shieldin depletion: co-depleting 53BP1 restored PALB2 focus formation to nearly control levels, RIF1 depletion had a weaker effect, whereas MAD2L2/REV7/REV7 or Shieldin depletion does not appear to significantly mask the defect caused by BRCA1 loss. These results fit well with the model of 53BP1 being the major chromatin anchor for the rest of the complex, with Shieldin binding to ssDNA via the OB-folds of FAM35A, and RIF1 and MAD2L2/REV7 bridging the chromatin- and the

ssDNA-interacting components (Greenberg, 2018). Importantly, the recruitment of RIF1, MAD2L2/REV7 and Shieldin depends on 53BP1 binding to chromatin adjacent to DSB, while 53BP1 focus formation is independent of the other factors (Dev et al., 2018; Noordermeer et al., 2018; Findlay et al., 2018). Finally, we would like to mention that reviewer 3 suggested a possibility of the “separation of function” between 53BP1 and Shieldin in regards to PALB2 loading, and this is exactly what our results indicate.

Reviewer 2

Loss of BRCA1 impairs HR and sensitizes cells to PARP inhibitor olaparib, a chemotherapeutic drug that has been approved to treat BRCA1/2-deficient ovarian, breast and other cancers. It has been reported that loss of 53BP1 can promote resistance of BRCA1-deficient cells to PARPi. In this manuscript, the authors investigated the mechanisms underlying this resistance. They reported that loss of 53BP1 partially rescues HR in BRCA1- but not PALB2- or BRCA2-deficient cells. They showed that depletion of 53BP1 rescues PALB2 focus formation in BRCA1-deficient cells and that this rescue is dependent upon the chromatin association motif (ChAM) of PALB2. They went on to further show that ChAM of PALB2 interacts with the acidic patch of H2A-H2B. They concluded that 53BP1 and RNF169 compete with PALB2 binding to chromatin and that loss of 53BP1 or RNF169 in BRCA1-deficient cells allows PALB2 better access to bind the surfaces (acidic patch) on nucleosomes. This rescue in PALB2 binding to chromatin contributes to the resistance of BRCA1- and 53BP1-depleted cells to PARPi. There are a number of major issues (below) with this manuscript and as a result, it is premature to be considered for publication in Nature Communications.

We thank Reviewer 2 for his/her comments and suggestions, which we address in detail below.

Major issues:

1. The concept that ChAM of PALB2 binds nucleosomes is not new. The authors claim that ChAM of PALB2 binds the acidic patch of H2A-H2B, however, the evidence supporting this claim is weak and insufficient. The authors need to show if there is a direct and physical interaction between ChAM and the acidic patch. The authors also need to show if such an interaction also exists with full-length PALB2.

While the ChAM domain has been reported to bind to chromatin, we show for the first time that the target for this domain is minimally a nucleosome core particle, mediated at least in part via the acidic patch. We have reconstituted this using fully purified, defined components (either generated in *E. coli* or chemically synthesised) *in vitro* with no other intermediary reagents as well as including relevant controls. By the very nature of the experiments performed, the interaction we have seen must be

a direct physical interaction. Furthermore, we draw to the reviewer's attention the fact that we have also strengthened our results with further repeats and mutations in our *in vitro* studies (Fig 4 g-h, and S5c of the revised manuscript).

Regarding NCP interaction with full length PALB2, we have performed pull-down experiments using full length GFP-PALB2 WT and Δ ChAM affinity-purified from 293T cells and recombinant NCPs, and can clearly see an interaction, not present with GFP alone control. However, both WT and Δ ChAM proteins appeared to bind NCP with equal affinity (data not shown). This could reflect the ability of full-length PALB2 to interact both with histones (via ChAM) and DNA (Nepomuceno et al., 2017; Deverishetty et al., 2019). We suggest that exploring such possibilities should be considered beyond the scope of our current manuscript.

2. The quality of the data presented in Fig. 4a is very poor. The recombinant MBP-ChAM protein seems to be degraded. Why is there a huge variation in the amount of H2B, H4 and Δ H4 in the input? The lack of Δ H4 NT in MPB-ChAM pull-down (last lane) seems to suggest that the N-terminal tail of H4 may be engaged in binding with ChAM, which would not support their claim that ChAM interacts with the acidic patch of H2A-H2B.

We thank this reviewer for pointing out these issues, and agree that the quality of this figure required improvement. Accordingly, we have now repeated the experiment and present the new data in our revised figure 4a, which support the same conclusions that we had drawn. Previously, the discrepancy in the loading and pull-down of histone H4 variants, full-length and Δ NT, was due to the binding preferences of the antibodies used, with lower recognition of the truncated H4 compared to the full-length protein. We have repeated this experiment using different antibodies and replaced the data in figure 4a.

3. In Fig. 4d, the authors showed that ChAM containing a single mutation of R414A or R421A is as defective as ChAM-4M in binding nucleosomes. The authors suggest that ChAM binds the acidic patch of H2A-H2B through electrostatic interaction. If so, does a conservative substitution of R414K or R421K affect ChAM interaction with

nucleosomes? Does a conservative substitution of R414K or R421K affect the ability of PALB2 to confer sensitivity to Olaparib, HR activity and RAD51 foci, etc?

We thank the reviewer for suggesting these experiments. Accordingly, we generated the conservative substitutions of R414 and R421 and thereby obtained some very revealing data (fig. 4h). In good agreement with our previous findings, it appears that the region around R414 is critical for interaction with nucleosomes (fig. 4h). In our revised discussion (pp.10-11), we propose this arginine might directly project into the acidic patch forming charged hydrogen bonds in a manner similar to other chromatin binding proteins (the “arginine anchor”). On the other hand, the effect of mutating R421 to alanine (R421A) on NCP binding is weaker, compared to R414A, whereas a ChAM with the conservative R421K substitution binds NCPs almost as effectively as the wild-type ChAM. These results suggest that, unlike structurally crucial R414, R421 is likely contributing to NCP binding by stabilising electrostatic interactions. Interestingly, mutations in this area described in cancer patients also seem to ablate interaction with nucleosomes (new fig. S5c).

4. In Fig. 3, the authors showed that PALB2 Δ ChAM does not confer olaparib sensitivity in the presence of BRCA1 but does so in the absence of BRCA1 and 53BP1. Previously it has been reported that PALB2 Δ ChAM does confer olaparib sensitivity in cells proficient for BRCA1 (Bleuyard et al. 2011). Although the authors made no mention of this discrepancy in the manuscript, they need to address it. Is their observation on PALB2 Δ ChAM and olaparib sensitivity unique to RPE-p53-KO cells? In addition, what is the effect of PALB2 Δ ChAM on HR activity as well as BRCA2 and RAD51 foci in the presence or absence of BRCA1 and 53BP1?

We thank the reviewer for bringing up this point. In the revised manuscript, we now mention the previously published data regarding olaparib sensitivity of BRCA1 proficient cells expressing Δ ChAM mutant (Bleuyard et al., 2011). We would like to note that the differences between our work and that of Bleuyard and colleagues might come both from using different cell lines and/or the fact that in the earlier study (Bleuyard et al., 2011), the expression levels of PALB2 Δ ChAM mutant tended to be lower in all the cell lines used.

5. In Fig. 2 and S2f, RPE1 Venus-PALB2 clone #1 was used in their analysis. However, Fig. S2b clearly showed that the clone #1 does not contain the expected 3.5-kb band that indicates Venus insertion. It is puzzling and concerning how the authors were able to show that this clone #1 behaves the same as the clone #15 that does have the Venus insertion.

We thank the reviewer for spotting this inconsistency. It indeed appears that in the PCR-based screening for Venus tag integration, the 3.5 kb band did not get amplified from the genomic DNA of clone #1 using primers P1 and P3 (new fig.S1A). We have used a different combination of primers, P2 and P3, to amplify across the targeted locus with P2 annealing just upstream of the gRNA-target site and P3 annealing downstream of the right homology arm (new fig. S2C). Notably, the original screen for Venus tag integration was performed simultaneously in two ways: by isolating gDNA and testing it by PCR as well as by high-throughput imaging of the cells. Clone #1 was selected in the imaging screen based on intact cell morphology and ability to form Venus-PALB2 damage-induced foci. We have also performed anti-GFP immunoprecipitations from both clone #1 and #15, and have confirmed that Venus-PALB2 from both clones interacted with BRCA2 and MRG15, the main constitutive partners of PALB2 as well as core histones, e.g. H2B (see below).

6. The authors claim that both 53BP1 and RNF169 are able to compete with PALB2 binding to chromatin. It has been reported that RNF169 limits 53BP1 deposition to DSBs although it does not completely abrogate 53BP1 at DSBs (An et al. 2018). Therefore, one might expect PALB2 focus count to be higher in BRCA1/53BP1-

deficient cells depleted for RNF169 compared to BRCA1/53BP1-deficient cells expressing siControl. Instead the authors showed that depletion of RNF169 did not affect PALB2 focus count in BRCA1/53BP1-deficient cells (Fig. 5c), which does not appear to support their claim.

In our original data, we found that while we consistently observed that RNF169 depletion had much stronger effects on increasing PALB2 foci in BRCA1/53BP1-proficient (fig. 5a-b) than in BRCA1/53BP1 KO (fig. 5c-d) cells, in the latter background, these effects were not statistically significant. However, upon repeating the experiment several more times to accumulate further data, we now establish statistically significant differences between the samples, as now reflected in our revised fig. 5c-d.

7. Based on their finding that depletion of RNF169 leads to an increase in PALB2 focus formation in WT cells, the authors suggest that RNF169 and PALB2 compete for their binding to the acidic patch of H2A-H2B. This conclusion is poorly supported and needs to be further substantiated. For example, RNF169 binds to the acidic patch through its LR-motif. Do mutations in LR-motif that abolish RNF169 binding to the acidic patch lead to an increase in PALB2 focus formation?

We thank the reviewer for this constructive comment. To address the question how RNF169 might interfere with PALB2 chromatin binding, we have generated stable cell lines with doxycycline-inducible expression of RFP-RNF169 WT or Δ LR, and tested the ability of Venus-PALB2 to form IRIF upon siRNA-mediated depletion of the endogenous protein and moderate overexpression of exogenous RNF169. The results are presented in the new fig. 5e,f,g. In brief, these data show that overexpression of WT RNF169 has a negative effect, while expressing RNF169 Δ LR has an overall positive effect, on PALB2 IRIF formation.

Other issues:

1. Lack of explanation of asterisks in many figures.
2. Figure legends need to be better described. For example, in Fig. 3c and 3e, what is the meaning of the number after WT, Δ ChAM, Δ MRG and Δ Ch/M?

3. Which of PALB2 Δ ChAM, Δ ChAM13 or Δ ChAM13, is used in clonogenic survival assays in Fig. 3f, 3g, 4e-4g?
4. Fig. 5d, the status of BRCA1 is not correctly indicated.

We thank the reviewer for these comments, which have all been addressed in our revised manuscript.

Reviewer #3

Overall, the data are well presented, the results clear and the model integrate all the data. My main concern is if the data as they are can really completely exclude an effect of 53BP1 on alleviating the mild defect in DNA resection known to be associated with BRCA1 deficiency. Even if such defect is too small to be visualized by RPA foci or FACs it cannot be excluded that it can contribute to the Rad51/PALB2 recruitment. Alternative ways to analyze this will be to use a method that quantify resection processivity or to repeat some critical experiments with KO of RIF1, REV7 or Shieldins. As the whole shieldin complex is required to inhibit resection, any effect on suppression of the BRCA1 resection defect should be equally observed in any of those KOs. However, on those KO cells, 53BP1 will still be attached to chromatin, likely blocking PALB2 binding through the ChAM domain. Thus, in this setup, a separation of function on resection/PALB2 recruitment should be observed.

We thank Reviewer 3 for his/her for positive and constructive comments, and apologise that our interpretation of the results was not coherent enough in our initial submission. We address this and other issues below.

The fact that the immunofluorescence (IF)-based techniques used to evaluate DNA end-resection are not very sensitive is acknowledged in our revised manuscript. BRCA1-depleted cells show the strongest defect in resection in the first 15-30 minutes (Cruz-Garcia et al., 2014). In our experiments, cells were processed for IF 5-8 hours after irradiation or for FACS 1 hour after addition of 1 μ M camptothecin; i.e. at the relatively late time points. At these later time points, we consistently observe minor differences in RPA signal intensity or focus count across all samples. Notably, our data are in good agreement with the results presented in fig. 2D of Polato et al. (2014), where DNA end-resection was tested by measuring BrdU signal following irradiation of BRCA1-deficient MEFs. Even though we do not see striking resection defect in BRCA1-deficient at the late time points, we consistently find defective RAD51 focus formation in those cells.

To address the question of the reviewer about the effect of Shieldin depletion on PALB2 focus formation in BRCA1-depleted cells, we have performed experiments in

which we co-depleted BRCA1 with various components of 53BP1-Shieldin complex. In order to evaluate PALB2 focus formation, these experiments had to be done via siRNA-mediated depletions of 53BP1, RIF1, MAD2L2/REV7 and Shieldin components in cells with endogenously Venus-tagged PALB2. The results of these experiments are presented in new fig. S8 and discussed on pp.11-12. In this set up we observed that the effect of PALB2 focus rescue in BRCA1-depleted cells was gradually “diluted” from 53BP to Shieldin depletion: co-depleting 53BP1 restores PALB2 focus formation to nearly control levels, RIF1 depletion has a weaker effect, whereas MAD2L2/REV7/REV7 or Shieldin depletion do not appear to significantly mask the defect mediated by BRCA1 loss. These results fit well with the model of 53BP1 being the major chromatin anchor for the rest of the complex with Shieldin binding directly to ssDNA via the OB-folds of FAM35A and RIF1 and MAD2L2/REV7 bridging the chromatin- and the ssDNA-interacting components (Greenberg, 2018). Importantly, the recruitment of RIF1, MAD2L2/REV7 and Shieldin depends on 53BP1 binding to chromatin adjacent to DSB, while 53BP1 focus formation is independent of the other factors (Dev et al., 2018; Noordermeer et al., 2018; Findlay et al., 2018). Therefore, in accord with the reviewer’s hypothesis, we do observe a “separation of function” between 53BP1 and Shieldin with regard to PALB2 loading.

Minor comments:

On Page 5, the laser line experiment is mentioned a couple of times as S3b, and it should be referred as S3c.

Page 5, end of second paragraph, a period is missing between “(Fig. 2c; supplementary movies)” and “Together”.

These minor issues have all been addressed in our revised manuscript.

Reviewer #4

This is an original and provocative study that gives more insight in the regulation of DNA DSB repair in 53BP1;BRCA1 double deficient cells. Similar to the recently published work by Zong et al. (reference 37), the authors investigated how loss of 53BP1 can restore RAD51 loading in BRCA1 deficient cells. They show that PALB2 recruitment to RPA-IRIF positive cells is increased in 53BP1; BRCA1 double deficient cells. In contrast to Zong et al, they claim that restoration of DNA end-resection does not play a major role in this process. This is a bold claim that requires more experimental evidence. The authors do not convincingly prove that the restoration of PALB2 recruitment is independent on end-resection. In addition, is not clear how 53BP1-loss-mediated increased PALB2 binding to the H2A-H2B acidic patch can lead to increased focal accumulation of PALB2 at sites of DNA damage. Nevertheless, the data in this study do suggests that 53BP1-loss-mediated restoration of HR in BRCA1 deficient cells goes beyond increased end-resection. This adds to our insight in the regulation of DNA double strand break repair, a process that is important for tumor suppression and treatment response. Specific comments.

We thank Reviewer 4 for his/her for positive and constructive comments.

1. Based on the data presented I am not convinced that BRCA1 'has a much weaker impact on resection than on RAD51 filament formation ..'. Data from several groups have confirmed the end-resection defects in BRCA1 deficient cells shown in reference 10 (Bunting 2010), including Drost et al. (2011 PMID: 22172724, 2016 PMID: 27454287) and He et al.(2018, PMID: 30464262). Discrepancies with the data in this manuscript and other findings may stem from the use of different experimental settings or techniques. E.g. measuring nuclear RPA intensity (Fig S1F) may not be a very sensitive approach. There should be stronger evidence that increased PALB2 recruitment to damaged DNA is not due to increased end-resection and by itself sufficient to restore HR (e.g. with the help of something like the PALB2-RNF8FHA fusion protein from reference 37). If the authors cannot provide more convincing experimental evidence, their conclusions should be adjusted. Independent of possible additional data, they should discuss their findings in relation to previous observations that do suggest a role for restoration of end-resection.

We apologise that our explanation of the results in the original manuscript was not clear enough; we have now amended the text in an attempt to make it clearer (p.5). We do not mean to suggest that we disagree with the conclusion that BRCA1 influences DNA-end resection; and upon detailed assessment, we believe that our data do not disagree with those in the papers mentioned by the reviewer. On the contrary, Drost et al (2016) demonstrate that cells carrying *BRCA1*^{185delAG} mutation can have reduced DNA end-resection but can still form RAD51 foci and perform HR, and this is what gives rise to olaparib resistance. Supposedly, those cells express low levels of RING-less BRCA1 protein (Drost et al., 2016). Importantly, the RING-less BRCA1 retains intact PALB2-binding capacity and therefore it can recruit BRCA2/RAD51 to DSBs, which agrees with recent study by Nacson et al. (2018) whereby it was established that BRCA1-deficient tumours can become resistant to PARP1 inhibition only when they express BRCA1 hypomorphs capable of interacting with PALB2. Regarding the end-resection, in the text of our original manuscript, we cited previously published papers in which it was clearly established that BRCA1 controls kinetics of DNA end-resection but is not absolutely required for the actual process (see a comprehensive review by Densham & Morris, 2019). In two of the studies the extent of resection in BRCA1-depleted cells was evaluated using the technique known as single molecule analysis of resection tracks (SMART), which allows accurate measurement of the physical length of ssDNA (Cruz-Garcia et al., 2014; Densham et al., 2016). Even though the tracks are shorter in BRCA1-depleted cells, their lengths are still measured in tens of microns encompassing tens of thousands of nucleotides, which would accommodate thousands of RPA complexes. Making use of this information, we explain why we detect RPA signal by the indirect immunofluorescence approach in our experiments. Moreover, we consistently image cells at 5-8 hours after irradiation, which is a relatively late time frame for accurate evaluation of the efficiency of DNA end-processing. At these later time points, we consistently observe minor differences in RPA signal intensity or focus count across all samples. Our data are in good agreement with the results presented in fig. 2D of Polato et al. (2014), where DNA end-resection was tested by measuring BrdU signal following irradiation of BRCA1-deficient MEFs. Although we do not see a striking resection defect in BRCA1-deficient at the late time points, we consistently find defective RAD51 focus formation in those cells.

2. It is not clear to me how increased binding of PALB2 to the H2A-H2B acid patch can increase focal accumulation at sites of DNA damage. The authors speculate that may be mediated by RNF168. PALB2 interacts with RNF168 via its WD40 domain, not via the Cham domain (Luijsterburg et al., PMID: 28240985). Does PALB2 Cham domain mutation then indirectly decrease the interaction with RNF168? Or does 53BP1 loss (also?) directly affect RNF168? The authors should at least discuss their findings in relation to the results from Luijsterburg et al.

We agree with the reviewer that our mentioning of the involvement of RNF168 in PALB2 focal recruitment was too brief. We have included a section addressing this in the discussion section of our revised manuscript (p.11). We think that PALB2, like most chromatin binding proteins described to date, is associating with chromatin in a multivalent manner. PALB2 likely forms stabilising interactions via the ChAM domain, the recently identified DNA binding domain (Deveryshetty et al., 2019) and other DDR factors, such as RNF168 (Lujsterburg et al., 2017) and RAD51C (Park J et al., 2014). While the ChAM domain is sufficient for interaction with chromatin, it seems that it is unlikely to mediate recruitment to sites of DSBs per se. This DSB specificity is likely brought about by binding to RNF168 (Luijsterburg et al., 2017), which is recruited specifically to its own DSB-induced chromatin mark. Like many other chromatin binding proteins, multiple separate interaction interfaces confer both specificity and affinity to ensure the appropriate response. A good example of this is provided by 53BP1, which interacts with both H4k20me2, H2Ak15ub and the nucleosome surface. Ablation of any one of these interaction interfaces is sufficient to greatly reduce both association *in vitro* and DNA-damage focus formation in cells.

Minor comments

1. In figure 1 the authors show that 53BP1 depletion does not rescue the HR defect of PALB2 or BRCA2 depleted cells. The latter is in line with the fact that loss of 53BP1 does not rescue proliferation of Brca2 deficient MEFs (reference 11, PMID: 20453858), which should be acknowledged.

We thank the reviewer for this comment. In our revised manuscript, we now mention the inability of 53bp1 depletion to rescue proliferation of Brca2 deficient MEFs (p.4).

2. It is not clear to me why it is emphasized on page 4 that 53BP1 inactivation rescues RAD51 IRIF formation in BRCA1 deficient RPE1 cells. This is a well know feature of 53BP1; BRCA1 double deficient cells (described in many papers, including the references 10 and 11) and also known for RPE1 (Durocher lab). If this is merely to validate their experimental system, this should be acknowledged.

We have amended the text accordingly (p.5).

3. For clarity it would be good to indicate in figures 3 and 4 (and not only in the legends) which graphs belong to 53BP1;BRCA1 proficient and deficient cells.

The figures have been modified accordingly (see new fig. 3 & 4)

Reviewers' comments:

Reviewer #1 (Remarks to the Author):

The revised paper shows convincingly that 53BP1 not only blocks resection in BRCA1 deficient cell as previous demonstrated, but also inhibits subsequent PALB2 chromatin binding via interaction with the H2A-H2B acidic patch. These results are entirely consistent with a recent manuscript (PMID: 31653568). Importantly, the latter did not address the role of the acidic patch which is the major focus of the current manuscript. Overall, the manuscript has improved considerably. Nevertheless, there are a few key issues that need to be resolved before publication:

#1. The authors argue that because overexpression of RNF169 stimulates SSA in HR-deficient cells and that HR normally suppresses SSA, it supports their finding that RNF169 competes with PALB2. This reasoning avoids the fundamental question of whether the authors detect more 53BP1 foci in RNF169-deficient cells, as reported by others. If the answer is yes, then why is 53BP1 not blocking PALB2 in this context? If the answer is no, what could be causing the discrepancy with multiple published studies? Notably, RPA32 staining appears to be weaker in siRNF169 cells that are wildtype for BRCA1, which would be consistent with increased 53BP1. It is therefore puzzling that PALB2 foci would also increase, as the central claim of this paper is that 53BP1 blocks PALB2. On the other hand, in BRCA1/53BP1 cells, it is more plausible that loss of RNF169 could increase PALB2, as there is no longer 53BP1 present. Altogether, the conclusion that RNF169 competes with PALB2 is not convincing.

The authors did show that overexpression of RNF169 reduced PALB2 foci. The effect is mild in BRCA1 proficient cells but strong in BRCA1/53BP1 cells. RNF169 overexpression has been reported to stimulate SSA. An alternative explanation could be that overexpression of RNF169 leads to over-resection and subsequent recruitment of RAD52, which in turn prevents PALB2 loading. Because this was not looked at, the current data cannot decisively prove that RNF169 itself competes with PALB2.

#2. The authors co-depleted BRCA1 and 53BP1/RIF1/REV7/Shieldin and observed that loss of 53BP1 imparts a strong rescue of PALB2, while RIF1 depletion has a minor effect and REV7/Shieldin depletion had no effect. However, this is not consistent with the authors' proposal that Shieldin can interfere with loading of RAD51 onto resected DNA. If this is true then depletion of Shieldin should have a major impact on PALB2 foci, which they do not observe. If REV7/Shieldin depletion has no impact on PALB2, how is RAD51 loaded and HR rescued in BRCA1/REV7 or BRCA1/Shieldin deficient cells? It seems likely that Shieldin also blocks PALB2/RAD51 loading in BRCA1 deficient cells via its ssDNA binding affinity, as recently published (PMID: 31653568). This could help their argument.

Reviewer #2 (Remarks to the Author):

Belotserkovskaya et al. have made efforts to improve the manuscript, but a few major issues from the previous round remain outstanding and should be addressed.

1) The authors did not address if a conservative substitution of R414K or R421K affect the ability of PALB2 to confer sensitivity to olaparib, HR activity and RAD51 foci (major issue #3 from the previous round). The authors showed that R414 is structurally crucial whereas R421 is likely contributing to NCP binding by stabilizing electrostatic interactions. Both R414 and R421 were mutated in the ChAM-4M mutant. Conceivably the olaparib sensitivity conferred by the ChAM-4M mutant in Fig. 4e might be caused by disruption in the ChAM structure rather than a defect in the ChAM-nucleosome interaction. The authors need to investigate if a single substitution of R414K alone confers olaparib sensitivity in RPE1 BRCA1/53BP1 KO cells. In addition, the authors need to

investigate if R421K is able to rescue olaparib sensitivity compared to R421A in RPE1 BRCA1/53BP1 KO cells.

2) The authors did not address if their observation on PALB2 Δ ChAM and olaparib sensitivity is unique to RPE-p53-KO cells (major issue #4). There is a discrepancy regarding the impact of overexpression of PALB2 Δ ChAM on olaparib sensitivity between their work and previously published work by Bleuyard et al. In addition, it is of importance to know if their observation on PALB2 Δ ChAM and olaparib sensitivity can be extended to other cell types, e.g. cancer cells. The authors should at least repeat their experiments described in Fig. 3f and 3g in a few other cell types to see if their observation is cell type specific.

3) Major issue #5, the authors should acknowledge in the text of the manuscript that they failed to amplify the 3.5 kb band for RPE1 Venus-PALB2 clone #1. They should also include their GFP coIP figure in the manuscript to support their claim for clone #1.

Minor comment:

Several figure panels are incorrectly quoted in their response to this reviewer, e.g. fig. 4h (does not exist in the manuscript), new fig. S1A and new fig. S2C.

Reviewer #3 (Remarks to the Author):

The authors have taken on board all my comments and suggestion and analyze the role of the rest of the Shieldin complex in PALB2 occlusion. Interestingly, they have seen a dilution of the effect, reinforcing the idea that 53BP1 has a effect that is mainly independent on the rest of the Shieldin complex.

Overall, I appreciate the effort the authors have made and gladly support the publication of the paper in Natc. Communication.

Reviewer #4 (Remarks to the Author):

The authors have improved the discussion of their data in relation to previously published work and also added further experimental evidence to support their claims. However, it remains unclear to me if HR restoration in 53BP1;BRCA1 deficient cells may at least partially be explained by effects on DNA end-resection. I agree that BRCA1 is not essential for this process, but end-resection defects are consistently observed in BRCA1 deficient cells. In my opinion, the additional data presented in figure S8 (in response to reviewer 3, who had the same concern) do not answer the question. Knockdown of RIF1, REV7 or SHLD2 indeed does not seem to have significant effects on PALB2 IRIF (although it seems that they are only compared with si53BP1, not with siCtrl) but also has minimal effects on RAD51 IRIF (in this experimental setting). Given the importance and the quality of this work I recommend publication, but there are a few points in the text that need to be adjusted:

1) Introduction page 3: "As loss of 53BP1 or its downstream effectors partially rescues the HR defect of cells lacking functional BRCA1, and important unresolved question is how this is achieved: does it act by rebalancing DNA end-resection, by restoring proficient RAD51 loading, or both?"

The discussion suggests a bit of both (with indirect effects on end-resection via RIF1, REV7 and SHLD1/2/3), but starts with emphasizing the importance of RAD51 loading via PALB2 without addressing if restored end-resection might increase PALB2 recruitment. The clear question in the introduction asks for an equally clear conclusion in the discussion.

2) Results page 5: "Collectively, our results are coherent with a recent report showing that 53BP1 loss can restore efficient RAD51 loading and HR in cells expressing hypomorphic BRCA1 derivatives that retain an ability to interact with PALB2 ..".

Although I understand that this "highlights the importance of recruiting PALB2 to DSBs in order to conduct HR", the results in this manuscript suggest that hypomorphic BRCA1 derivatives are not essential for increased HR in the absence of 53BP1, which is in contrast to what Nacson et al. (ref 13) suggest. This apparent discrepancy may be explained by different experimental settings, and – as discussed on page 11 - Zong et al. (ref 43) also show that there is BRCA1-independent PALB2 recruitment (via RNF168). As this manuscript does not analyze PALB2 recruitment in the context of hypomorphic BRCA1, confusion may be avoided by some rephrasing . Alternatively, a brief discussion of the apparent discrepancy is required.

Reviewer #1 (Remarks to the Author):

We thank this reviewer for his/her insightful and constructive comments. Please see our responses to the issues raised below.

The revised paper shows convincingly that 53BP1 not only blocks resection in BRCA1 deficient cell as previously demonstrated, but also inhibits subsequent PALB2 chromatin binding via interaction with the H2A-H2B acidic patch. These results are entirely consistent with a recent manuscript (PMID: 31653568). Importantly, the latter did not address the role of the acidic patch which is the major focus of the current manuscript. Overall, the manuscript has improved considerably. Nevertheless, there are a few key issues that need to be resolved before publication:

#1. The authors argue that because overexpression of RNF169 stimulates SSA in HR-deficient cells and that HR normally suppresses SSA, it supports their finding that RNF169 competes with PALB2. This reasoning avoids the fundamental question of whether the authors detect more 53BP1 foci in RNF169-deficient cells, as reported by others. If the answer is yes, then why is 53BP1 not blocking PALB2 in this context? If the answer is no, what could be causing the discrepancy with multiple published studies?

As the reviewer points out, it has indeed previously been reported that the overall 53BP1 IRIF count goes up in RNF169-depleted cells. As we are focusing on homologous recombination (HR)-relevant events, such as DSB resection, PALB2 and RAD51 loading, all of which take place in S/G2 cells, we quantified 53BP1 foci in RNF169-depleted cells co-stained for the markers of S/G2 cell cycle stage (CENPF and PCNA). We found that in S/G2 (CENPF/PCNA-positive) cells, 53BP1 focus count increases moderately (up to 20%); however, in G1 (CENPF/PCNA-negative) cells, the number of 53BP1 foci almost doubled compared to the control cells (see graph below). Given the above findings and the fact that in BRCA1-proficient cells, 53BP1 should be efficiently displaced from DSBs destined for HR (Chapman et al., 2012 J Cell Sci; Densham et al., 2016 NSMB; Densham & Morris, 2019 Front Mol Biosci), we propose that RNF169 depletion will have a minor impact on 53BP1 IRIF-mediated interference with HR in S/G2 cells. We have revised our text to clarify these points (page 10), and include the data below in our revised Fig S6c.

Notably, RPA32 staining appears to be weaker in siRNF169 cells that are wildtype for BRCA1, which would be consistent with increased 53BP1. It is therefore puzzling that PALB2 foci would also increase, as the central claim of this paper is that 53BP1 blocks PALB2. On the other hand, in BRCA1/53BP1 cells, it is more plausible that loss of RNF169 could increase PALB2, as there is no longer 53BP1 present. Altogether, the conclusion that RNF169 competes with PALB2 is not convincing.

The reviewer points out that RPA32 foci appear to be weaker in RNF169-depleted BRCA1/53BP1-proficient cells. We apologize if the image presented in figure 5a gave somewhat misleading view of RPA32 IRIF. We have replaced the original image with another, more representative, field of cells where RPA32 foci in siRNF169-depleted cells appear comparable to those in other samples (new figure 5a). Please see below the graph with quantification of RPA32 foci/cell at 6 hours after irradiation with 5Gy. The overall RPA32 IRIF counts obtained via high-throughput high-content imaging are somewhat higher in both RNF168- and RNF169-depleted BRCA1/53BP1-proficient cells relative to siCTRL-treated cells, which is in agreement with previously postulated altered regulation of HR processes in cells depleted for RNF168 or RNF169.

The authors did show that overexpression of RNF169 reduced PALB2 foci. The effect is mild in BRCA1 proficient cells but strong in BRCA1/53BP1 cells. RNF169 overexpression has been reported to stimulate SSA. An alternative explanation could be that overexpression of RNF169 leads to over-resection and subsequent recruitment of RAD52, which in turn prevents PALB2 loading. Because this was not looked at, the current data cannot decisively prove that RNF169 itself competes with PALB2.

We agree with the reviewer that we have not looked directly at RAD52 recruitment in our setting and thus, even though all our data strongly support PALB2/RNF169 competition for chromatin binding, we cannot rule out effects mediated via the involvement of RAD52-mediated SSA. We have amended the subheading of our manuscript by changing it from "RNF169 competes with PALB2 chromatin binding" to "RNF169 counteracts PALB2 chromatin recruitment" (page 9). We have also added the discussion of RNF169-related data to the 'Discussion' section (page 12), in which a possibility of RAD52 and SSA is now clearly mentioned.

#2. The authors co-depleted BRCA1 and 53BP1/RIF1/REV7/Shieldin and observed that loss of 53BP1 imparts a strong rescue of PALB2, while RIF1 depletion has a minor effect and REV7/Shieldin depletion had no effect. However, this is not consistent with the authors' proposal that Shieldin can interfere with loading of RAD51 onto resected DNA. If this is true then depletion of Shieldin should have a major impact on PALB2 foci, which they do not observe. If REV7/Shieldin depletion has no impact on PALB2, how is RAD51 loaded and HR rescued in BRCA1/REV7 or BRCA1/Shieldin deficient cells? It seems likely that Shieldin also blocks PALB2/RAD51 loading in BRCA1 deficient cells via its ssDNA binding affinity, as recently published (PMID: 31653568). This could help their argument.

We thank the reviewer for helping us better explain our findings regarding PALB2 focus formation upon depletion of different components of 53BP1-Shieldin axis in light of the recent publication (PMID: 31653568). Following the advice of the reviewer, we have now extended the relevant part of our discussion and mentioned the suggested publication (page 14). In addition, we also refer to another article published while our manuscript was in revision (PMID: 31645724), which strongly supports our findings regarding the effects of depletion of different components of 53BP1-Shieldin axis on PALB2 and RAD51 focus formation: specifically, by means of super resolution microscopy, Ochs et al. (Nature, October 2019) demonstrated that only depletion of 53BP1 or RIF1, but not of Shieldin, leads to disruption of chromatin domains next to DSBs and aberrant spreading of DNA repair proteins. These latter findings could explain why we detect stronger PALB2 accumulation in 53BP1- and (to a lesser extent) in RIF1-depleted but very weak PALB2 IRIF recovery in REV7- or Shieldin-depleted BRCA1-deficient cells.

Reviewer #2 (Remarks to the Author):

Belotserkovskaya et al. have made efforts to improve the manuscript, but a few major issues from the previous round remain outstanding and should be addressed.

We thank this reviewer for acknowledging that our manuscript has been improved.

1) The authors did not address if a conservative substitution of R414K or R421K affect the ability of PALB2 to confer sensitivity to olaparib, HR activity and RAD51 foci (major issue #3 from the previous round). The authors showed that R414 is structurally crucial whereas R421 is likely contributing to NCP binding by stabilizing electrostatic interactions. Both R414 and R421 were mutated in the ChAM-4M mutant. Conceivably the olaparib sensitivity conferred by the ChAM-4M mutant in Fig. 4e might be caused by disruption in the ChAM structure rather than a defect in the ChAM-nucleosome interaction. The authors need to investigate if a single substitution of R414K alone confers olaparib sensitivity in RPE1 BRCA1/53BP1 KO cells. In addition, the authors need to investigate if R421K is able to rescue olaparib sensitivity compared to R421A in RPE1 BRCA1/53BP1 KO cells.

We agree with the reviewer that we have only partially addressed his/her original comment #3, i.e. we performed an extensive site-directed mutagenesis across the ChAM domain of PALB2 to identify key residues that contribute to the domain's interaction with the nucleosome core particle (NCP). We found that substitution of Arg414 to either Ala, Lys, or Gln ablates ChAM-NCP binding, supporting the notion that R414 could be an "arginine anchor", a feature common to proteins that bind the H2A-H2B acidic patch. Once we had obtained the above data, integrating single point mutants into cells, validating the ensuing cell lines and then performing functional studies on them could unfortunately not be done in the time allocated for the revision. Moreover, we think that to get the most comprehensive conclusions regarding the effects of single point mutations, these types of experiments should be combined with structural studies, which are clearly outside the scope of the current manuscript.

2) The authors did not address if their observation on PALB2 Δ ChAM and olaparib sensitivity is unique to RPE-p53-KO cells (major issue #4). There is a discrepancy regarding the impact of overexpression of PALB2 Δ ChAM on olaparib sensitivity between their work and previously published work by Bleuyard et al. In addition, it is of importance to know if their observation on PALB2 Δ ChAM and olaparib sensitivity can be extended to other cell types, e.g. cancer cells. The authors should at least repeat their experiments described in Fig. 3f and 3g in a few other cell types to see if their observation is cell type specific.

Although we agree that reproducing findings in several different cell types might be of added value to support our model, we would like to point out that the scale of the requested experiments is incompatible with the revision time-frame. The discrepancies between the results reported by Bleuyard et al. and our results could be due to difference in expression levels of PALB2. In the former study (Bleuyard et al., 2012), the expression level of PALB2 Δ ChAM was significantly lower than that of the PALB2 WT (for expression levels, please compare fig.2B of Bleuyard et al. and our Fig. S4c; and please note that in both cases, the same cell lines were later used for respective sensitivity/viability assays). We had highlighted this issue in our original rebuttal letter.

3) Major issue #5, the authors should acknowledge in the text of the manuscript that they failed to amplify the 3.5 kb band for RPE1 Venus-PALB2 clone #1. They should also include their GFP coIP figure in the manuscript to support their claim for clone #1.

The data that the reviewer is referring to are presented in figure S2b. As we were unable to amplify the fragment of genomic DNA in targeted cells using primers annealing on either side of the homology arms, we performed additional PCR using primers, one of which anneals inside the tagging cassette and another downstream of the right homology arm, i.e. in the body of the gene, and could confirm the successful Venus-tag integration (new fig. S2e). The annealing site for the primer upstream of the left homology arm is located 1.7kb upstream of the first exon of PALB2 and a possible mutation of this primer site does not affect Venus-PALB2 expression in clone #1, as is evident from IRIF formation and the co-IPs, which, following the request by the reviewer, we have now included in a new fig. S2f. The relevant information can be found in the modified figure legend for fig. S2.

Minor comment:

Several figure panels are incorrectly quoted in their response to this reviewer, e.g. fig. 4h (does not exist in the manuscript), new fig. S1A and new fig. S2C.

We apologize for mistakes in figure referencing in the Response letter: fig. 4h is indeed not there; fig. S1a should have been "S2a".

Reviewer #3 (Remarks to the Author):

The authors have taken on board all my comments and suggestion and analyze the role of the rest of the Shieldin complex in PALB2 occlusion. Interestingly, they have seen a dilution of the effect, reinforcing the idea that 53BP1 has a effect that is mainly independent on the rest of the Shieldin complex.

Overall, I appreciate the effort the authors have made and gladly support the publication of the paper in Natc. Communication.

We thank this reviewer for his/her positive comments and for supporting publication of our manuscript in Nature Communications.

Reviewer #4 (Remarks to the Author):

We thank this reviewer for recommending our manuscript for publication in Nature Communications.

The authors have improved the discussion of their data in relation to previously published work and also added further experimental evidence to support their claims. However, it remains unclear to me if HR restoration in 53BP1;BRCA1 deficient cells may at least partially be explained by effects on DNA end-resection. I agree that BRCA1 is not essential for this process, but end-resection defects are consistently observed in BRCA1 deficient cells. In my opinion, the additional data presented in figure S8 (in response to reviewer 3, who had the same concern) do not answer the question. Knockdown of RIF1, REV7 or SHLD2 indeed does not seem to have significant effects on PALB2 IRIF (although it seems that they are only compared with si53BP1, not with siCtrl) but also has minimal effects on RAD51 IRIF (in this experimental setting). Given the importance and the quality of this work I recommend publication, but there are a few points in the text that need to be adjusted:

1) Introduction page 3: “As loss of 53BP1 or its downstream effectors partially rescues the HR defect of cells lacking functional BRCA1, and important unresolved question is how this is achieved: does it act by rebalancing DNA end-resection, by restoring proficient RAD51 loading, or both?” The discussion suggests a bit of both (with indirect effects on end-resection via RIF1, REV7 and SHLD1/2/3), but starts with emphasizing the importance of RAD51 loading via PALB2 without addressing if restored end-resection might increase PALB2 recruitment. The clear question in the introduction asks for an equally clear conclusion in the discussion.

We agree with the reviewer that this is an important point. Thus, we have re-written the first paragraph of the discussion (page 10) to link it up with the question stated in the introduction.

2) Results page 5: “Collectively, our results are coherent with a recent report showing that 53BP1 loss can restore efficient RAD51 loading and HR in cells expressing hypomorphic BRCA1 derivatives that retain an ability to interact with PALB2 ..”. Although I understand that this “highlights the importance of recruiting PALB2 to DSBs in order to conduct HR”, the results in this manuscript suggest that hypomorphic BRCA1 derivatives are not essential for increased HR in the absence of 53BP1, which is in contrast to what Nacson et al. (ref 13) suggest. This apparent discrepancy may be explained by different experimental settings, and – as discussed on page 11 - Zong et al. (ref 43) also show that there is BRCA1-independent PALB2 recruitment (via RNF168). As this manuscript does not analyze PALB2 recruitment in the context of hypomorphic BRCA1, confusion may be avoided by some rephrasing. Alternatively, a brief discussion of the apparent discrepancy is required.

We thank the reviewer for this comment. To avoid any confusion, we have removed the mentioning of hypomorphic BRCA1 from the text.

REVIEWERS' COMMENTS:

Reviewer #1 (Remarks to the Author):

The major result of this paper is that 53BP1 competes with PALB2 chromatin binding, thereby inhibiting HR in BRCA1 deficient cells. The argue that RNF169 also competes with PALB2. Although perhaps it CAN compete, the physiological relevance is unclear since loss of RNF169 does not have a role in rescuing HR in BRCA1 deficient cells. Moreover, 53BP1 foci increase in RNF169 deficient cells (albeit less dramatically in S/G2), making it unclear what impact RN169 has. Thus, I feel it would be confusing to include data on RNF169, which will distract from and dilute the main conclusions.